



# Ambient conditions prevailing during hail events in central Europe

Michael Kunz[1,2], Jan Wandel[1], Elody Fluck[1,*], Sven Baumstark[1,+], Susanna Mohr[1,2], and
Sebastian Schemm[3]

[1]Institute of Meteorology and Climate Research (IMK), Karlsruhe Institute of Technology (KIT), Karlsruhe, Germany
[2]Center for Disaster Management and Risk Reduction Technology, KIT, Germany
[3]Institute for Atmospheric and Climate Science, ETH Zürich, Zürich, Switzerland
[*]now at: Department of Earth and Planetary Sciences, Weizmann Institute of Science, Rehovot, Israel
[+]now at: Heine + Jud, Stuttgart, Germany

**Correspondence:** Michael Kunz (michael.kunz@kit.edu)

**Abstract.** Around 26,000 severe convective storm tracks between 2005 and 2014 have been estimated from 2D radar reflectivity for parts of Europe, including Germany, France, Belgium, and Luxembourg. This event-set was further combined with eyewitness reports, convection-related parameters from ERA-Interim reanalysis and synoptic-scale fronts based on the same reanalysis. Our analyses reveal that about a quarter of all severe thunderstorms in the investigation area were associated with

a front. Over complex terrains, such as in southern Germany, the proportion of frontal convective storms is around 10–15%, while over flat terrain half of the events require a front to trigger convection. Frontal hailstorms on average produce larger hailstones and have a longer track. These events usually develop in a high-shear environment. Using composites of environmental conditions centered around the hailstorm tracks, we found that dynamical proxies such as deep-layer shear or storm-relative helicity become important when separating hail diameters and, in particular, their lengths; 0-3 km helicity as a dynamical proxy

performs better compared to wind shear for the separation. In contrast, thermodynamical proxies such as Lifted Index or lapse rate show only small differences between the different intensity classes.

## 1 Introduction

Severe convective storms (SCS) are responsible for almost one-third of total damage by natural hazards in Germany and central Europe (Re, 2019). Examples of recent major loss events include the two supercells on 27/28 July 2013 related to the depression

*Andreas* with economic losses of EUR 3.6 billion mainly due to large hail (Kunz et al., 2018) or storm clusters during *Ela* on 8-10 July 2014 with economic losses of EUR 2.6 billion (SwissRe, 2015) caused by both large hail and severe wind gusts (Mathias et al., 2017). Given the major damage associated with SCS, particularly due to large hail, there is a considerable and increasing need to better understand the local probability of SCS, their intensity and relation to prevailing atmospheric precursors.

Several authors have attempted to establish relations between SCS/hailstorms and favorable atmospheric environments (for Europe: Manzato, 2005; Groenemeijer and van Delden, 2007; Kunz, 2007; Sánchez et al., 2009; Mohr and Kunz, 2013; Púčik et al., 2015; Sánchez et al., 2017; Madonna et al., 2018, among others). Hail-conductive environments have been estimated either from proximity soundings or from model data/reanalysis, both available over several decades and, depending on the



spatial resolution, on a regional, continental, or global scale. According to Púčik et al. (2015), for example, large hail most likely
occurs for high values of CAPE and bulk wind shear. While the former is directly related to the intensity of the updraft, the latter
is decisive for the organization's form of the convective systems – single cells, multicells, supercells, mesoscale convective
systems (MCS) (Markowski and Richardson, 2010). In addition, several studies have suggested that SCS preferentially occur
during specific weather regimes, such as European blocking or teleconnection patterns (Aran et al., 2011; García-Ortega et al.,
2011; Kapsch et al., 2012; Piper et al., 2019). However, to date, no study has investigated environmental conditions according
to hailstone size and footprint, despite their relevance to overall storm damage.

Forecast experience has shown that synoptic fronts, particularly cold fronts during the summer months, can significantly
modify the convective environment, primarily due to increasing convective available energy (CAPE) and decreasing convective
inhibition (CIN) in combination with cross-frontal circulations leading to lifting and enhanced vertical wind shear. By combin-
ing hailstorm tracks determined from radar data over Switzerland between 2002 and 2013 with front detections (Schemm et al.,
2015) based on the Consortium for Small Scale Modelling (COSMO) analysis, Schemm et al. (2016) found that up to 45%
of storms in northeast and southern Switzerland were associated with a cold front. They concluded that mainly wind-sheared
environments created by the fronts provide favorable conditions for hailstorms in the absence of topographic forcing.

Difficulties in analyzing environmental conditions prior to or during hailstorms usually arise from insufficient direct hail
observations that may serve as ground truth. The number of ground weather stations is too small to reliably detect all SCS.
High-density hailpad networks exist in only a few regions across Europe (e.g., Merino et al., 2014; Hermida et al., 2015)
and therefore cannot be used to reproduce entire hailstorm footprints. In order to compensate for this monitoring gap, remote
sensing instruments such as satellite (Bedka, 2011; Punge et al., 2017; Ni et al., 2017; Mroz et al., 2017), lightning (Chronis
et al., 2015; Wapler, 2017), or radar (Holleman et al., 2000; Puskeiler et al., 2016; Nisi et al., 2018) due to their area-wide
observability are used to estimate the frequency and intensity of SCS. In particular, weather radars can give some indications
of hail occurrence using either radar reflectivity above a certain threshold (e.g., Mason, 1971; Hohl et al., 2002) or at specific
elevations in combination with different height specifications (melting level, -20° C environmental temperature, top of the
storm cell; Waldvogel et al., 1979; Smart and Alberty, 1985; Witt et al., 1998). While observations by dual-polarization radars
offer better predictions for hail (e.g., Heinselman and Ryzhkov, 2006; Ryzhkov et al., 2013; Ryzhkov and Zrnic, 2019) these
systems have been installed in Europe only recently and cannot be used for climatological studies.

Another important data source for hail is severe weather reports from trained storm spotters or eye-witnesses that are pooled
into severe weather archives such as the European Severe Weather Database (ESWD; Dotzek et al., 2009). Although reporting
is selective and biased towards population density and available spotters, these reports provide valuable information about the
intensity of the various convective phenomena associated with SCS such as maximum hail diameter. The combination of these
reports with storm tracks estimated from radar observations allows us to reconstruct entire footprints of SCS and/or hailstorms.

In our study, we have reconstructed SCS tracks from two-dimensional (2D) radar reflectivity using a cell tracking algorithm
during a 10-year period (2005–2014) over central Europe including France, Germany, Belgium, and Luxembourg. Since high
radar reflectivity does not guarantee to have hail on the ground (mainly because of potential melting hailstones and the relation
$Z \sim D^6$, where $D$ is the hail size diameter), we created an additional subsample by filtering the radar-derived tracks with





ESWD hail reports. This step not only ensures that the subsample consists of hailstorms (HS) solely but also merges hailstorm

tracks and maximum hail diameters. Afterward, we investigate characteristics and environmental conditions at the time and location of the events unfolding for different classes of hail diameter, track lengths (lifetime) and the relationship with synoptic-scale fronts. Environmental conditions are assessed by constructing composites of meteorological fields from ERA-Interim reanalysis centered around the location of a single storm.

The main scientific questions of our study are the following:

– How frequent are SCS associated with a front? Do the characteristics of frontal SCS differ from those without a front?

– How do the environmental conditions in terms of thermodynamical and dynamical parameters differ between hail diameter classes, track lengths, and frontal/non-frontal events?

– Does the propagation direction of hailstorms depend on hail size?

The paper is structured as follows: Section 2 introduces the datasets and methods used. Section 3 deals with the frequency of

SCS and HS and Section 4 examines the role of synoptic cold fronts and convective storms. Section 5 statistically investigates environmental conditions prevailing around the storms for different classes of hail size and track length. Section 6 synthesizes and summarizes the major findings, while the most important conclusions are drawn in Section 7.

## 2    Data and Methods

The investigation area is central Europe, including Germany, France, Belgium, and Luxembourg, from 2005 to 2014, where

data were available. Since SCS in Europe occur mainly in the summer half-year (SHY; Berthet et al., 2011; Punge and Kunz, 2016), all analyses refer to the period from April to September.

### 2.1    ESWD hail reports

The ESWD, managed and maintained by the European Severe Storms Laboratory (ESSL), is the only multinational database and by far the largest archive of hail reports in Europe. Quality-checked reports of SCS and related phenomena originate from

storm chasers and trained spotters, sometimes supplemented by newspaper reports. In our study, we consider hail reports of all quality levels (70.4% of all reports were confirmed, 29.0% were at least plausibility checked). This includes both large hail with a diameter of at least 2 cm usually given in increments of 1 cm (in rare cases of 0.5 cm) and hail layers with a depth of at least 10 cm, regardless of hail diameter. In those cases, and when a hail size is not specified (usually in case of small hail), the diameter is set to 1 cm.

During the 10-year investigation period, a total of 4,577 reports of severe hail in the study area is available. Most reports stem from Germany (76.5%), followed by France (21.1%), Belgium (1.7%), and Luxembourg (0.7%). This distribution does not reflect the occurrence probability of SCS but is primarily due to ESSL originally being a German initiative.





Because of the large spatial extent of the study area in a west-east direction, we converted the time stamps for the daily cycle analysis (only for that; cf. Fig. 2) from UTC into Local time (LT) by adding $\Delta t = 24\text{h}/360°\text{lat} = 4$ min per degree starting from $0°$ lat.

## 2.2 Reanalyses

Atmospheric conditions prevailing over a larger area around the SCS tracks are studied using ERA-Interim (Dee et al., 2011) reanalysis from the European Center for Medium-Range Forecast (ECMWF). This data set, which was also used for the detection of synoptic cold fronts (see Sect. 2.3), is represented as spherical harmonics at a T255 spectral resolution (approx. 80 km) on 60 vertical levels from the surface up to 0.1 hPa with a temporal resolution of 6 hours.

Mesoscale environments of the hailstorm tracks are characterized by severe storm predictors representing both thermodynamical and dynamical conditions. We tested and applied several convection-related parameters, but focus here only on those proxies with the highest prediction skill: Surface Lifted Index (SLI) representing latent instability (Galway, 1956), Deep-Layer Shear (DLS) as the difference of the wind vectors between 500 hPa and the surface, and 0–3 km Storm-Relative Helicity (SRH) quantified by:

$$\text{SRH} = \int (\mathbf{v}_h - \mathbf{c}) \cdot (\nabla \times \mathbf{v}_h)\, dz \tag{1}$$

$$= \int \left[ -(u - c_x)\left(\frac{\partial v}{\partial z}\right) + (v - c_y)\left(\frac{\partial u}{\partial z}\right) \right] dz \tag{2}$$

where $\mathbf{v}_h = (u, v)$ is the horizontal wind vector and $\mathbf{c} = (c_x, c_y)$ the (constant) cell motion vector. Helicity is a measure of the degree to which the direction of motion is aligned with the (horizontal) vorticity of the environment $\omega_h = \nabla \times \mathbf{v}_h$ (Markowski and Richardson, 2010). Only streamwise vorticity, which is a prerequisite for supercells bearing the largest hailstones, contributes to SRH (Thompson et al., 2007). As the convective cell tracking algorithm directly computes $\mathbf{c}$ for each SCS/HS event (see next Sect. 2.4), we used these values to quantify SRH from ERA-Interim without relying on a semi-empirical relation such as that from Bunkers et al. (2000).

## 2.3 Cold front detection

Synoptic-scale cold fronts are detected in ERA-Interim based on the method outlined in Schemm et al. (2015), which is briefly summarized here. To identify and locate fronts in the reanalysis, we used the thermal front parameter (TFP; Renard and Clarke, 1965; Hewson, 1998) defined as

$$\text{TFP} = -\nabla|\nabla \theta_e| \cdot \frac{\nabla \theta_e}{|\nabla \theta_e|}, \tag{3}$$

where $\theta_e$ denotes the equivalent potential temperature at 850 hPa, a widely-used choice in the forecasting community. The first term in Eq. 3 represents the gradient of the frontal zone ($|\nabla \theta_e|$), which must be higher than $4\,\text{K}\,(100\,\text{km})^{-1}$. The second term is the unit vector of the $\theta_e$ gradient. The TFP hence captures changes of the gradient of the frontal zone along the gradient itself. The frontal zone is strongest where $\text{TFP} = 0$, and its leading-edge is where $\text{TFP} = \text{Max}$. For the detection of propagating





synoptic-scale fronts, a minimum length of $500\,\mathrm{km}$ and a minimum advection speed of $3\,\mathrm{m\,s^{-1}}$ is defined (Schemm et al., 2015). These two thresholds sufficiently remove land-sea contrast and thermal boundaries from Alpine pumping from the

dataset and limit the data to fronts typically associated with extratropical cyclones.

## 2.4    Radar data and storm tracking

Tracks of SCS are identified from two-dimensional (2D) radar reflectivity based on the precipitation scan at low elevation angles. Radar data with a spatial and temporal resolution of 1 km and 5 min, respectively, were provided by Meteo France and by German Weather Service (DWD) as entire radar composites. Whereas all 17 German radars operate in C-band, 19 radars

in France are C-band and five each are S-band and X-band. The area in France covered by the S-band radars is rather small ($< 5\%$ of the total area) compared to that captured by C-band, and mainly restricted to the southwest (S-band radars at Opoul, Nîmes, Bollene, and Collobrieres). Because of the dominance of C-band radars, we did not distinguish between the two radar types. X-band radars, exclusively operating in the Maritime Alps in southeast France, are not considered due to their strong attenuation of the radar signal.

Storm tracks were reconstructed by applying a modified version of the cell tracking algorithm TRACE3D originally designed for 3D reflectivity in spherical coordinates (Handwerker, 2002). Thus, TRACE3D has to be modified to rely on 2D radar reflectivity in Cartesian coordinates (Fluck, 2017). The tracking algorithm first identifies all convective cells (reflectivity core, RC) embedded into larger "regions of intense precipitation" (ROIP; Handwerker, 2002). Afterward, the weighted center (barycenter) of all RCs is tracked spatially over subsequent time intervals $dt$ by establishing a temporal connection between

the detected RCs. For each RC, a 2D shift velocity vector $\mathbf{v}_T$ is calculated in different ways, depending on whether and over what distance an RC has already been detected in previous scans. The new position of the RC is estimated from $\mathbf{s}_T = \mathbf{v}_T \cdot \delta t$ within a certain search radius $r$, which depends on the length of $\mathbf{s}_T$ and the distance to the closest neighboring RC. This process is repeated for all subsequent scans until the complete track of a convective cell is reconstructed. The algorithm considers different processes such as cell splitting or merging. Correction algorithms are implemented for undesired radar effects such

as the bright band or anomalous propagation (so-called Anaprop).

We considered only storm tracks with a lower threshold of $Z \geq 55\,\mathrm{dBZ}$, referred to as the Mason (1971) criterion. Several studies have provided evidence that suggests this threshold to provide sufficient skill for hail detection (e.g., Holleman et al., 2000; Hohl et al., 2002; Kunz and Kugel, 2015; Puskeiler et al., 2016). However, it cannot be excluded that the sample also contains individual storm tracks with heavy rainfall, but without hail. In addition, we eliminated all single grid points with high

radar reflectivity ($\geq 55\,\mathrm{dBZ}$), but without lightning within a radius of 10 km. This filter is based on the assumption that SCS are always accompanied by lightning. Note that the filter only eliminates single spurious signals but keeps the tracks that are composed of numerous radar grid points.

During the investigation period, 26,012 SCS tracks were identified, each containing the following parameters: center (lat/lon) of the track including date and time, mean angle, width, total length and duration; the latter two quantities allowing us to

compute the storm motion vectors $\boldsymbol{c}$ required for SRH (cf. Eq. 1). For further details on the tracking and the results, see the study by Fluck (2017).





## 2.5 Combination of SCS tracks with other parameters

### 2.5.1 Combination with fronts

To match the SCS tracks with synoptic front detections (cf. Sect. 2.3), we first compute the minimum horizontal distance $d_i$ between the two events:

$$d_i = \sqrt{(a_i \cdot \cos(lat \cdot 2\pi/360) \cdot l)^2 + (b_i \cdot l)^2}, \tag{4}$$

where $a_i$ is the longitudinal distance between a frontal grid point $i$ and the grid points of an individual storm track, $b_i$ is the same for the latitude, $lat$ is the position (latitude) of the storm track, and $l$ is the (constant) distance of $1°$ latitude ($\approx 111.32\,\text{km}$). The cos function in the equation takes into account the poleward convergence of the lines of longitude. For each front detection, we compute the distance $d_i$ to all grid points defining the track of an SCS identified in the same 6-hour period. The minimum of all $d_i$, thus $d_{min} = \min(d_i)$ defines the minimum distance between the front and the related SCS.

Frontal SCS are defined as those events where a front is located within a search radius of $R = L/2 + 200\,\text{km}$ ($L$ is the length of an SCS track) around the storm track, i.e. when $d_i < R$. Assuming a front to act as a potential trigger for convection, the distance between the two events must be limited (Trapp, 2013). For this reason and because of the low temporal and spatial resolution of the front detections, we set the constant part of $R$ to $200\,\text{km}$. Note that changing this part to a value of 300 or $400\,\text{km}$ has no significant effect on the results. The constant part in $R$ ($L/2$) considers only the time of the center of the SCS for the synchronization between the two events. The longer $L$, the larger the temporal and spatial difference between tracks and fronts can be and, thus, the larger $R$ must be.

To account for temporal coincidence, we consider the timestamp of the SCS centers that must be within the period of the front detections (00, 06, 12, 18 UTC). When the SCS center is exactly between the ERA-Interim run times (03, 09, 15, 21 UTC), both time frames are used in the calculations of $d_i$. Since the front detections are available for six-hour intervals only, the time difference between the centers of the SCS and the fronts is at most three hours. Considering the start time of the SCS instead of that at the center has only a small marginal effect on the results because of both the low temporal resolution of the reanaysis and the comparatively short duration of the SCS tracks (exponential distribution; 73% of all SCS have a duration of 2 hours and less).

### 2.5.2 Combination with ESWD data

The SCS tracks derived from the radar composites are additionally combined with the ESWD reports to assign each track a maximum hail diameter. This is done by considering both the date/time and the horizontal distance $d_i$ between a certain track and the nearest ESWD report in the same way as described above for the fronts. Only ESWD reports with $d_{min} \leq 10\,\text{km}$ to the closest grid point are considered; these storms are hereafter referred to as hailstorm (HS) tracks. A tolerance of $10\,\text{km}$ is necessary for two reasons: In some cases, the ESWD reports do not give an exact position; and hailstones falling to the ground may drift with the horizontal wind over distances of several kilometers (Schuster et al., 2006). When an ESWD report coincides with several tracks, we further considered the time of the report if specified. Still unclear cases (around 100 events



corresponding to 2% of all cases) were not considered in the event set. If more than one ESWD report is assigned to a single
storm track, we considered only that with the maximum reported hail diameter.

The combination of SCS tracks with ESWD reports substantially reduced the sample size to 985 HS tracks. The main reasons
for the much low number of HS compared to SCS events are an underreporting of hail events, especially over France, and some
SCS tracks accompanied by heavy rain and not by hail. Nevertheless, this sample size is still sufficient for the investigation of
environmental conditions for different intensity classes.

For all investigations, we separated the maximum hailstone diameter into three different classes (samples): $D < 3\,\mathrm{cm}$ (48,0%
of all HS tracks), $3 \leq D \leq 4.5\,\mathrm{cm}$ (37.0%), and $D \geq 5\,\mathrm{cm}$ (15.0%).

### 2.5.3 Composite construction

The investigation of the environmental conditions around the HS tracks is based on composites of convection-related pa-
rameters from ERA-Interim. The effect of latitudinal dependence on the horizontal difference between the grid points in the
reanalysis is considered by transferring the latter to Cartesian coordinates with a grid resolution of approximately $50\,\mathrm{km}$. The
composites are constructed from mean values of different convective parameters (cf. Sect. 2.2) obtained by moving spatial
windows of 800 km in latitude and longitude at the center of the SCS/HS tracks (i.e., $\pm 400$ km to the north, south, east, and
west from the center of the track). As mentioned above, using the start location instead of the center does not affect the results
because of the limited spatial extent of the tracks (mean lengths of frontal/non-frontal HS tracks are 56.8 and 96.2 km, respec-
tively). In addition, due to the low resolution of the ERA-Interim data, it can be assumed that the convective environment is
not modified by ongoing convective storms.

The single ERA-Interim fields with a size of $800 \times 800\,\mathrm{km}^2$ are averaged either for all events or for different categories of
events related to hail diameter classes, HS track lengths, and frontal vs. non-frontal HS events. Since most of the HS events
propagate from south-southwest to north-northeast, we have not aligned the fields accordingly. Note, however, that according
to a test where this was realized, the results remained essentially the same.

Temporal coincidence is ensured by using the reanalysis fields with the smallest time difference to the HS events. Therefore,
the largest time difference between the environmental conditions and the HS events is 3 hours.

## 3 Frequency of SCS and HS

### 3.1 Spatial distribution of SCS and HS events

The frequency of both SCS and HS events shows a large spatial variability with an overall increase from north-to-south, i.e.,
with the distance to the Atlantic. Embedded in this large-scale trend are several regional-scale hotspots such as those over and
downstream of the Massif Central (France) or downstream of the Black Forest (southwest Germany). Over France, however,
SCS occur much more frequently than in Germany (Fig. 1). By contrast, HS events are fare more frequently detected over
Germany as a consequence of the substantial underreporting in France and alluded to previously. Despite these differences in





the detection efficiency, there is a clear relationship between the two records: regions with an increased SCS frequency also show an increased HS frequency and vice versa.

## 3.2 Daily and seasonal cycle

Both HS and SCS events (the latter not shown) feature pronounced seasonal and diurnal cycles with a maximum in the afternoon in the warmest months of July and August. While the number of HS is lowest in April and September and dominated by smaller-
sized hail, the months of May to July are similar with the highest number of HS events of the diameter class $D \geq 5\,\mathrm{cm}$ in June (Fig. 2a). A comparison of the three summer months shows that events with large hail are rarest in July. Reasons for this counterintuitive result might be a decrease in frontal events, which have low hail sizes on average (cf. Sect. 2.5.1), or reduced reporting in this month due to summer holidays.

The diurnal cycle is much more pronounced than the seasonal cycle. The minimum number of HS events occurs in the early
morning hours between 03 and 09 LT and the maximum in the afternoon between 15 and 18 LT (Fig. 2b). The largest increase occurs between 12 and 15 LT and the largest decrease after 21 LT. A total of 841 events, which correspond to 85.4% of the HS sample, is registered in the period from 12 to 21 LT.

A separation of the diurnal cycle according to the hail diameter shows that during the first half of the day (00–12 LT), most events are associated with hail smaller than 5 cm. Especially from 03 to 09 LT, hailstones are the smallest of the entire day.
This result, however, must be treated with care because of the low number of events in that period (26 events) in combination with the potential underreporting by spotters in the night. During noon and afternoon, the proportion of hail with a diameter of at least 5 cm increases, and the highest probability of occurrence is between 15 and 18 LT. In the evening and night (18–00 LT), the relative proportion of large hail remains almost constant.

The pronounced diurnal cycle of the HS probability is closely linked to the warming of near-surface layers of air and the
associated increase in lapse rate / CAPE together with a decrease in CIN (Markowski and Richardson, 2010). In addition, triggering mechanisms such as low-level flow convergence in the wake of thermally-induced circulation over complex terrain or inhomogeneities in land cover are also connected to the diurnal temperature cycle. Studies using lightning data found similar diurnal cycles for most of the area except for the Mediterranean (e.g., Wapler, 2013; Piper and Kunz, 2017).

## 4 SCS and fronts

## 4.1 Front detections

The investigation area is frequently affected by synoptic-scale cold fronts (cf. Sect. 2.3). The number of fronts per grid point of the size $1° \times 1°$ during the 10-year investigation period ranges between 85 in eastern Germany and 175 near the Pyrenees (Fig. 3). Overall, front density in France is larger than in Germany.

During their propagation, cold fronts tend to weaken mainly because of friction in the lowest layers and horizontal mixing
of air mass properties. Usually, they also dissolve when the air from the warm sector has entirely lifted (occlusion). As the





largest fraction of fronts affecting central Europe propagates in east- to south-east directions, their detectable density gradually decreases in the same direction. In addition, an elevated front density can be found on the western and northern side (upstream) of large mountains such as the Pyrenees, Massif Central, and the Alps. These large mountain ranges tend to slow down the propagation of fronts, leading to an elevated frequency upstream when counting the time steps where a front prevails and not the individual fronts as is done here (Schemm et al., 2016). Thus, slow propagating fronts may be repeatedly detected and counted during the time steps of ERA-Interim (6 hours). In contrast, fronts occur less frequently downstream of larger mountains as well as at a greater distance to the sea, where the increasing continentality acts to weaken or even dissolve the fronts.

### 4.2 Frontal SCS and HS tracks

To assess the role of synoptic cold fronts in the probability and properties of SCS, we first discuss the spatial distribution of the ratio of frontal SCS relative to all SCS events. This ratio is computed independently for each single grid point with a size of $0.5° \times 0.5°$. Averaged over the entire area of Germany and over the 10-year study period, 18.9% of all SCS tracks are related to a cold front; for France, the ratio is slightly higher with 22.4% (Fig. 4). The most conspicuous feature in the spatial distribution of the frontal streaks is the strong gradient in the south-to-north direction, particularly over Germany. For example, while in the German northeast (Mecklenburg Lake Plateau) the share of frontal SCS reaches the highest value of 50%, it decreases to less than 10% in southern Germany over the Black Forest (SW Germany) and the region south of Nuremberg (SE Germany). In France, the spatial distribution of frontal SCS is smoother than in Germany. Most obvious here is an extended maximum of around 45% northeast of the domain's center and several minima with only a few percent near the coasts of both the North Atlantic and the Mediterranean.

If we compare the proportion of frontal SCS both with both the distribution of all SCS tracks (cf. Fig. 1) and with the frontal density (cf. Fig. 3), the opposite behaviour is often observed. In several regions with an increased number of fronts and/or SCS events, the number of frontal SCS is low and vice versa. This is especially true for Germany, but also for parts of France. Over complex terrain such as in SW Germany (Black Forest) or Southern France (Massif Central), where frontal SCS are comparatively rare, it can be assumed that orographically-induced vertical lifting is often sufficient to trigger convection so that a front is not necessary.

Considering HS instead of SCS events, we found that an even higher number, namely 25% of all HS tracks across the entire study domain are connected to a synoptic cold front. Because of the small number of HS track detections, especially in France (cf. Fig. 1), we do not show this relation here. Note, however, that if only areas with a sufficient number of events are considered, the spatial distributions of frontal HS and SCS tracks are quite similar.

For the HS events, a relation is found between the length of the tracks as detected by the radar algorithm and the maximum observed hail diameter (Fig. 5a). While the mean diameter for a length of $L < 50\,\mathrm{km}$ is around $2\,\mathrm{cm}$, it increases to around $3\,\mathrm{cm}$ for $50 \geq L < 150\,\mathrm{km}$ and to $4\,\mathrm{cm}$ for $L \geq 150\,\mathrm{km}$. Furthermore, the distributions of both quantities, maximum diameters and track lengths, differ between frontal and non-frontal steaks. Mean diameters are $3.3\,\mathrm{cm}$ in for frontal events and $2.73\,\mathrm{cm}$ for the others (Fig. 5b, left part). For hail size diameter classes of $< 2$, $2–3.5$, $4–5.5$, and $\geq 6$ cm, the ratio between frontal



and non-frontal events is 16.7, 23.1, 35.8, and 34.7%, respectively (not shown; note that the finer classes are used only in this
example). This means, the higher the probability of a nearby front, the larger is the hailstone diameter on average.

Differences between frontal and non-frontal HS events are also found for the length and mean propagation direction of the
tracks. While frontal HS has a mean length of 96.2 km (interquartile range of 40–125 km), non-frontal tracks are are almost
half shorter with 56.8 km (25–65 km; Fig. 5b, right part). Non-frontal HS events have a mean propagation angle of 215°
(interquartile range 185°–255°), whereas those connected to a front propagate slightly more to the east with a direction of
232° (interquartile range 217°–258°; not shown. Note that also the interquartile range ×1.5 is almost twice for non-frontal
compared to frontal HS). In that latter range of angles, also the largest hailstones can be observed.

These results suggest that fronts create hail-conducive conditions mainly through two effects: along-front advection of
moisture at lower levels leading to larger CAPE, and higher wind speed aloft enhancing vertical wind shear. The latter creates
an environment that favors the development of organized, more persistent thunderstorms such as multicells and supercells,
which plausibly explain the larger diameters and track lengths for HS events connected to a front compared to those without
a front. Furthermore, fronts also largely determine the orientation of the tracks, which results from the predominating general
flow direction from the west sector. Differences in prevailing environmental conditions are investigated in the next section.

## 5    Environmental conditions of HS events

Environmental conditions prevailing during HS events are investigated using SLI, DLS, and SRH from ERA-Interim reanalysis
(Section 2.2). The composites presented in this section show the mean fields of the respective proxy around the center of the HS
tracks (see Sect. 2.5.3). To examine environmental conditions depending on the intensity of the HS events, we further divided
the HS sample into 9 subsamples according to the observed hail diameter $D$ ($< 3$ cm, 3–4 cm, $\geq 5$ cm) and track length $L$
($< 50$ km, 50–100 km, $> 100$ km). The thresholds defining the classes were set so that each sample has a sufficient number of
events, although the sample sizes decrease from small to large diameters and lengths (see Table 1).

Further subdivision, for example the time of occurrence, was not carried out. Although scientifically interesting, this would
further reduce the sample sizes, particularly the most interesting high-intensity classes.

### 5.1    Mean Composites

Averaged over all classes of HS events, the SLI around the center of the tracks has a value of -3.8 K in the mean (Fig. 6a),
indicating a high potential for convective storms (Kunz, 2007). SLI has its absolute minimum about 140 km southeast of the
events, but the difference to the center, on average of 0.2 K, is almost negligible. Overall, a significant increase in convection
favoring conditions can be observed from the northwest of the HS center to the southeast. While these conditions prevail over
400 km to the south and east of the center, the area to the north and west sees higher and positive values of SLI, thus stable
conditions, at approximately 100–200 km distance already. The SLI field occurs rather smooth, mainly because of the low
resolution of ERA-Interim (a high-resolution reanalysis data set basically gives the same results, but shows a higher spatial
variability; not shown).


The vertical wind shear (DLS) has its maximum about $250\,\mathrm{km}$ to the west of the HS centers in an upstream direction (Fig. 6b). This spatial difference is plausible because, as alluded to previous, a trough usually prevails to the west of the events. Since DLS is dominated by the wind speed aloft ($500\,\mathrm{hPa}$), a trough manifests itself by a maximum in DLS. Considering the magnitude of DLS, it is found that the values are quite low with a mean of $12.5\,\mathrm{m\,s^{-1}}$ around the HS events. Several authors
have shown that organized convection capable of producing larger hail develops only in sheared environments above around $10\,\mathrm{m\,s^{-1}}$ (e.g., Weisman and Klemp, 1982; Markowski and Richardson, 2010; Dennis and Kumjian, 2017). This is one of the reasons to further subdivide the whole sample as mentioned above and shown in the next paragraph.

### 5.2 Composites for different classes of hail diameter and track length

Separating the hail events according to their intensity allows a detailed view of the prevailing environmental conditions. The
SLI composites show a slight decrease (higher instability) around the center of the HS events from small hail with shorter tracks ($\mathrm{SLI} \approx -3.7°C$) to large hail with longer tracks ($\mathrm{SLI} \approx -4{,}5°C$; Fig. 7). The strongest decrease in stability occurs for increasing hail diameter, while the composites are less sensitive to variations in track lengths. In all cases, the lowest instability prevails to the southeast of the hail events as was already found for the mean composite (cf. Fig. 6). Despite favorable environments for SCS, which predominate all classes, the highest instability in the case of larger hail is an indicator for higher
updraft speed within the thunderstorm cloud, which is a prerequisite for the growth to large hailstones.

The distance between the location of the events and the location of the highest instability is greater for longer tracks than for shorter ones, but only in case of small to medium-sized hail. At this point one may speculate that the reason for this shift might be related to the role of cold fronts, considering that longer tracks and larger hailstones are more often connected to a cold front as discussed in the previous section (cf. Fig. 5). The role of cold fronts versus environmental conditions will be investigated in
the next section.

In contrast to the thermodynamical proxy SLI, the dynamical quantity DLS shows significantly pronounced differences between the nine HS categories (Fig. 8). Even though DLS also distinguishes between the diameter classes, the largest differences are found for the three-length classes. For example, DLS has a mean value of $17\,\mathrm{m\,s^{-1}}$ for long tracks in the smallest diameter class ($D < 3\,\mathrm{cm}$), which is almost twice as high compared to short tracks with the same diameter class ($8{,}5\,\mathrm{m\,s^{-1}}$; Fig. 8, upper
row). The same applies to the other diameter classes. For long tracks with large hail, DLS reaches values of about $20\,\mathrm{m\,s^{-1}}$ and is thus in the range of the values given in the literature (e.g., Weisman and Klemp, 1982; Thompson et al., 2007; Markowski and Richardson, 2010). The area of the highest DLS values is located several hundred kilometers to the west of the HS events on average. For large hail, the DLS maxima are even higher and further away from the HS events. These events are usually triggered by upper-level troughs to the west, associated with higher wind speed at mid-troposphere levels. One may argue that
a relationship between DLS and track length prevail per se since both are dominated by the wind speed aloft. Note, however, that the separation of DLS applies not only to track length, but also to storm duration (not shown here, but see Wandel, 2017).

In addition to DLS, SRH has been suggested by several authors (e.g., Thompson et al., 2007; Kunz et al., 2018) to be an important proxy not only for the prediction of tornadoes but also for large hail. In our composite analyses, SRH (Fig. 9) shows even more pronounced differences between the nine HS categories compared to DLS. Hail events with shorter tracks on average



are in a range between 0 and $50\,\mathrm{m^2\,s^{-2}}$, while longer tracks have much higher mean values between 84 and $116\,\mathrm{m^2\,s^{-2}}$. In
addition, there is also an increase in SRH from small to large hail, which is weaker compared to the trend in the length classes.

Interestingly, the highest SRH values occur directly at or near the location of the hail event and not on the upstream side as
was the case for DLS.

### 5.3   Frontal vs. non-frontal HS

As already discussed in Section 4.2, the characteristics of HS tracks having a front nearby substantially differ from non-
frontal events, especially with regard to the maximum hail size and the track lengths (cf. Fig. 5). This suggests that prevailing
environmental conditions may likewise differ for the two kinds of events. Therefore, we additionally divided the HS sample into
frontal and non-frontal types. To ensure that enough events enter the subsamples, we made a further separation by considering
only two classes of hail diameters $D \lesseqgtr 3\,\mathrm{cm}$ and track lengths $L \lesseqgtr 75\,\mathrm{km}$.

Whereas the mean SLI composites are almost similar for frontal / non-frontal events (not shown), DLS shows significant
differences between the four classes (Fig. 11). Overall, DLS reaches higher values with larger gradients for frontal compared
to non-frontal events (Fig. 11, left vs. right column). However, when considering additionally the track lengths, much larger
differences in DLS can be found, but only for non-frontal events (Fig. 11, right column). While short tracks form at a DLS
of $10.9\,\mathrm{m\,s^{-1}}$ on average, long tracks require medium sheared environments, here with values of $15.9\,\mathrm{m\,s^{-1}}$. A similar result
is obtained for small hail sizes ($D < 3\,\mathrm{cm}$) with DLS even rising from 9.0 to $16.7\,\mathrm{m\,s^{-1}}$ (not shown). Furthermore, while the
DLS maximum for non-frontal events is located to the west of the center, it is more northwest for frontal events at a distance of
about $200\,\mathrm{km}$. Since almost all synoptic fronts in Europe propagate in a west-to-east direction, this location is a clear indication
that frontal HS events preferably develop in prefrontal environments (and not postfrontal).

### 5.4   Statistical evaluation of appropriate predictors

To further investigate which of the dynamical parameter, SRH or DLS, best distinguishes the HS intensity, we consider only the
two categories that correspond to the highest and lowest damage potentials: smaller hail with $D < 3\,\mathrm{cm}$ combined with short
track length of $L < 50\,\mathrm{km}$ and large hail with $D \geq 5\,\mathrm{cm}$ combined with longer tracks of more than $100\,\mathrm{km}$ (high-intensity
events). Frontal and non-frontal events are not separated. Environmental conditions are computed by the mean of the $3 \times 3$
ERA-Interim grid points centered around the HS locations.

Overall, the scatter plots presented in Figure 10 show a much clearer separation between the events when SRH is considered
(Fig. 10a) instead of DLS (Fig. 10b). About 50% of the high-intensity events have values of $100\,\mathrm{m^2\,s^{-2}}$ or greater for SRH,
while only 3% of the less-intensity events display these values. Furthermore, most of the latter events have values between -50
and $50\,\mathrm{m^2\,s^{-2}}$. It can also be seen that SLI for all events in these two categories vary between 0 and -10 K, with only a few
exceptions having positive values. Approximately 70% of the high-intensity events have values of -2.5 K or less. Unlike DLS
(Fig. 10b), splitting the events into two different categories is not possible. Even if most of the high-intensity events form in an
environment with DLS of at least $15\,\mathrm{m\,s^{-1}}$ (approx. 60% of these events), there are still many low-intensity events for larger
DLS values.




In addition, we quantitatively evaluated the prediction skill of certain parameter combinations using categorical verification (Wilks, 2011). Highest skill scores are given for SRH in combination with a thermodynamic proxy such as SLI or 700–500 hPa lapse rate. For example, the Heidke Skill Score (HSS), which is particularly suitable for predicting rare events, is 0.50 for the SRH–SLI combination (0.50 for lapse rate instead of SLI), but only 0.27 for the DLS–SLI combination (0.33 for lapse rate). This finding suggests considering SRH at least in addition to DLS when predicting the severity of SCS.

## 5.5 Differences in wind direction

Ordinary thunderstorms (single cells) usually propagate with the horizontal wind at mid-tropospheric levels, whereas supercells can deviate substantially from that, mainly because of induced vertical pressure deviations (Markowski and Richardson, 2010). To investigate whether such a relation can also be found in samples, we quantified the difference between the storm motion vector **c** of the HS events obtained from the radar tracking algorithm and the wind direction in 500 hPa from ERA-Interim using again the $3 \times 3$ grid point around the HS centers. Positive differences indicate right-moving supercells, negative values left-moving cells.

Most of the events with smaller hail ($D < 3\,\mathrm{cm}$) propagate approximately parallel to the wind vectors in 500 hPa; the mean difference between the tracks and the wind vectors is only 8° (Fig. 12a). About 13% of all HS events have a deviation between 30 and 60° to the right, while only 6% of the events show deviations to the left for this interval (-30 to -60°). Hail events with maximum diameters between 3 and 4.5 cm already show a deviation of the propagation direction preferably to the right of the wind vectors (Fig. 12b). 23% of all HS events propagate with the wind in 500 hPa (decreasing by 8% compared to small hail), while 38% of the tracks show a deviation between 10 and 30°.

HS events of the largest hail class not only show an increased spread of the propagation deviation but also the entire histogram is shifted to more right-movers (median: 17°; Fig. 12c). An angle difference between 10 and 30° is observed for 35% of all events. The largest difference to the other hail size classes is the comparatively high number of HS events between 30 and 60° (21%). In contrast, 27% of the events propagate with the wind in 500 hPa, and only 10% have a negative deviation to the left of the wind in 500 hPa. In summary, the larger the hailstone diameters, the stronger is the deviation of the cell's propagation direction from the flow at 500 hPa.

## 6 Discussion

Severe convective storms, chiefly hailstorms, are high-frequent perils that, due to their local-scale nature, affect only small areas (Changnon, 1977). Thus, their reconstruction requires high-resolution such as radar data as in our study. The results of the analyses show high spatial variability of both SCS and HS events, with a gradual increase with growing distance from the ocean and several hotspots, mainly over and downstream of mountain ranges. For example, as shown by Kunz and Puskeiler (2010), these hotspots are connected to flow convergence at lower layers in the low Froude number regime, when the flow tends to go around rather than over the mountains. Overall, the spatial distribution of SCS/HS events agrees with other studies on that topic considering different data sets such a combination of radar data with weather stations (Junghänel et al., 2016),



3D radar reflectivity (Puskeiler et al., 2016; Lukach et al., 2017), a longer period (Kunz and Kugel, 2015), or overshooting top detections from satellite (Bedka, 2011; Punge et al., 2017). This applies also to the seasonal and diurnal cycles (Nisi et al., 2016; Punge and Kunz, 2016; Nisi et al., 2018). The good quantitative and qualitative agreement is a strong indication of the reliability of our methods and results.

     All composites of environment parameters created for radar-derived HS tracks show a similar spatial pattern: whereas the
thermodynamic proxies such as SLI have their highest values at some ten kilometers up to 100 km southeast of the center of the HS events, the maxima of the dynamic proxies (DLS and SRH) are found to the northwest at a distance of 100 to 200 km. This applies to all intensity classes and to all proxies originally considered in our study (also for KO-Index and lapse rate, but not for PW, where the maximum is located north of the events).

     In total 651 of all 985 HS events have a southwest to the northeast propagation direction, reflecting the mean flow direction
at mid-troposphere levels. On average, HS events usually occur downstream of the eastern flank of a mid-troposphere trough, where southerly to southwesterly winds lead to the advection of unstable, warm, and moist air masses from the Mediterranean (Piper et al., 2019). The trough, on the other hand, creates an environment with increased wind shear and large-scale lifting. The axis of the trough is usually located several hundreds of kilometers upstream of the HS events, which explains why the highest shear is found on the western flank at larger distances. Furthermore, as convection initiation requires an additional
lifting mechanism to overcome the convective inhibition in the planetary boundary layer, the area downstream of a trough is an ideal location for the development of (organized) convection as shown, for example, by Wapler and James (2015) or Piper et al. (2019).

     The separation of the environmental composites into different classes of hail diameter and track length yields several interesting results. Thermal instability, as expressed for example by SLI, increases slightly (smaller values of SLI) from small hail
with shorter tracks to large hail with longer tracks, as might be expected. While the strongest decrease is found for increasing hail sizes, the composites are only marginally sensitive to variations in the track length. By contrast, the separation for DLS and SRH is much stronger, particularly for the track lengths. This dependence of the track lengths to DLS or SRH can be explained plausibly by the storm's organization. Low-to-medium sheared environments ($\sim 10\,\mathrm{m\,s^{-1}}$) permit single cells to develop (Markowski and Richardson, 2010), which are not able to produce large hail. For organized convective storms such
as multicells, supercells, or MCS, substantial shear ($\sim 20\,\mathrm{m\,s^{-1}}$) is required, which spatially separates the updraft from the downdraft. Supercell thunderstorms, bearing the largest hailstones, preferably develop in environments with DLS exceeding $18\,\mathrm{m\,s^{-1}}$ (Weisman and Klemp, 1982; Markowski and Richardson, 2010). High-resolution model simulations by Dennis and Kumjian (2017) show that increased DLS upstream elongates the storm's updraft downshear, providing an increased volume of the hailstone growth region, an increased hailstone residence time within the updraft, and a larger region for potential hail
embryos. Altogether, these mechanisms lead to increased hail masses and, thus, increased hail diameters, even though the average value of DLS for our event set is at the lower end of the typical value range for multicellular convection.

     The hypothesis that supercells preferably enter the subsample of long tracks / large hail is also supported by the findings of the differences between the propagation vector of cells determined by the tracking algorithm and the mean wind at 500 hPa from ERA-Interim reanalysis. The larger the hailstones, the larger is the relative share of events with a deviation mostly to the





right of the ambient wind. Because of vertical dynamic pressure perturbations, supercells tend to deviate substantially from the mean wind direction (Markowski and Richardson, 2010). So-called right-moving supercells, usually persisting after cell splitting (Klemp, 1987) because of positive linear dynamic forcing, may deviate from the mean wind direction by angles of up to 30°. Such deviations have already been observed for single supercells in Germany (Kunz et al., 2018). In contrast, multicell thunderstorms or MCS bearing smaller hailstones show fewer deviations from mid-tropospheric winds.

When a synoptic cold front is involved, the pre-convective environment usually changes on short time scales. During the summer months, cold fronts can substantially modify the pre-convective environment because of four independent effects: (i) lapse rate increase by cold air advection aloft; (ii) vertical lifting by frontal cross-circulations, which simultaneously increases CAPE and reduces CIN; (iii) increase of low-level moisture and, thus, of CAPE; and (vi) enhanced curvature of the hodograph related to the thermal wind equation (Markowski and Richardson, 2010). The latter, not directly connected to a front, potentially occurs several hundred kilometers upstream. Because of the above-listed factors, hail events associated with cold fronts are likely to have different properties than non-frontal events. We found, for example, frontal HS events to produce larger hail and longer tracks compared to non-frontal HS events on average.

The share of frontal SCS (and HS) to all events substantially varies among the regions. For example, whereas only a limited number of SCS in southern Germany have a front nearby, almost half of the events over northern Germany are front-related. By combining radar-based hail events for Switzerland between 2002 and 2013 with cold front detections (Schemm et al., 2015) based on COSMO analysis, Schemm et al. (2016) found that locally up to 45% of all hail events in northeastern and southern Switzerland are associated with a cold front. This is similar to our study region, where we identified values of up to 50% locally.

Over complex terrain, it can be assumed that moisture flux convergence at low levels caused by flow deviations at obstacles and local wind systems is the most important trigger mechanism for convection initiation (Weckwerth and Parsons, 2005; Barthlott et al., 2011; Trefalt et al., 2018). In contrast over mainly flat terrain such as in northern Germany, a front is often required as a trigger for convection. Instability and vertical wind shear are two additional effects that partly determine the probability of frontal SCS. These two quantities on average are highest in the southern parts of France and Germany, where frontal SCS are not very frequent (not shown). Thus, we conclude that the share of frontal SCS to all events is the result of the interaction of various influencing factors, mainly of thermal instability and lifting mechanisms to initiate convection.

When a front is nearby, HS events tend to develop east of the maximum of wind shear and northwest of the most unstable stratification. In contrast, non-frontal HS events frequently occur in proximity to the highest wind shear and most unstable conditions. In low-sheared environments, hailstorms capable of producing hail larger than 3 cm develop only when the air mass is highly unstable. Higher instability, in general, enables stronger updrafts that are required for the development of larger hailstones. For frontal HS events, the stratification remains almost the same, but with the highest instability located more to the southeast of the events. This region of highest instability, however, is characterized by lower shear. At the same time, assuming a trough prevailing at the western side of the HS events, large-scale descent associated with high-pressure systems tend to suppress convection initiation (Piper and Kunz, 2017). This relation also explains why the dynamical and thermodynamical conditions in terms of DLS and SLI prevailing during HS events for the different classes are consistent among themselves.





## 7 Conclusions

In our study, we have reconstructed a large number of past severe convective storms and investigated prevailing environmental conditions over a 10-year period in central Europe. The combination of SCS tracks derived from 2D radar data with hail reports from ESWD gave additional information on the hailstone size of a storm, but also ensured that the resulting subsample consisted of hailstorms solely. The resulting HS subsample allowed us to investigate prevailing environmental conditions from reanalysis as a function of hail size and track length. In addition, we have investigated how and through which mechanisms synoptic cold fronts modify the characteristics and the frequency of SCS / HS events. Our study is the first of its kind that relies on both hail size and track length, a combination essential for the damage potential of severe hailstorms.

The main conclusions from our research are the following:

- The probability of occurrence of both SCS and HS events shows a distinct seasonality with most of the events occurring between May and July, and a pronounced diurnal cycle with a maximum in the afternoon between 15 and 18 LT, where storms with large hail ($D \geq 5\,\mathrm{cm}$) are most likely to occur.

- Approximately one-quarter of all SCS across the investigation area is connected to a front, being usually pre-frontal events. Over complex terrain, such as in southern Germany, the share of frontal SCS is low (partly below 10%), while over flat terrain a front is more often required (up to 50% of all events) to trigger convection.

- Frontal HS events on average produce larger hailstones and have longer tracks. These events preferably develop in a high-shear environment related to the cold front.

- Dynamical proxies such as DLS or SRH become important when separating between hailstorms of different intensity classes (diameter and length), particularly when considering their length (or likewise their duration). Thermodynamic proxies such as SLI or lapse rate show only small differences around the event's centers between the different classes.

- SRH (0-3 km ) as dynamical proxy performs better compared to DLS when separating HS events according to hail size and track length.

- The larger the hail size, the larger the deviation between track direction and direction of the mean wind at $500\,\mathrm{hPa}$. Most of the large hail events ($D \geq 5\,\mathrm{cm}$) propagate to the right of the mean wind, suggesting an increased probability of right-moving supercells in that subsample.

A potential weakness of our study is that it relies on eye-witness reports (ESWD), which are biased towards denser populated regions and towards daytime. This constraint reduces not only the size of the HS sample but also creates a spatial bias as can be seen in the substantially lower number of HS events in France than in Germany. Furthermore, the estimation of the largest diameter for hailstones that may substantially deviate from a sphere creates additional uncertainty.

Despite the different sources of uncertainty and the limited representativity of the reports for several regions, the comparatively large sample including approximately 1000 events enables reliable statistical analyses when aggregated over the





whole investigation area. Furthermore, ESWD reports are the only dataset that gives additional information about hail diame-
ter. Insurance loss data used in several hail-related studies (e.g., Vinet, 2001; Schuster et al., 2006; Kunz, 2007) or data from
hailpad networks (e.g., Dessens and Fraile, 1994; Sánchez et al., 2017) cannot be applied because of the large spread inherent
in the damage–to–diameter relation or the limited regions gauged. In the future, ground-truth observations collected through
crowdsourcing via specific platforms such as the European Weather Observer App (EWOBS; Groenemeijer et al., 2017) or the
MeteoSwiss App (Trefalt et al., 2018; Barras et al., 2019) might overcome the underreporting of hail events.

The main findings and conclusions of our study can be considered in several ways. Above all, the results can (and should) be
implemented in the forecasting of SCS for lead times between 1 and 12 hours. This time range is of considerable importance
for many users as well as for issuing warnings of SCS/HS associated with high impact weather phenomena such as hail,
heavy rainfall, or severe wind gusts. In the hierarchy of prediction models, this time range is covered by nowcasting tools
and very short-range forecasts (Nisi et al., 2014; James et al., 2018). Hence, convective indices, particularly SRH or DLS,
might be employed in both systems. Our results can help to distinguish between less severe and more severe convection. When
focusing on the most severe storms, the magnitude and temporal evolution of SRH/DLS, and whether a front is nearby should
be considered. Finally, because there is evidence of an increase in the number of extremely strong weather fronts during the
summer over Europe (Schemm et al., 2017), our findings have implications for explaining trends and regional-scale variability
of front-related SCS and HS.

*Data availability.* 2D radar data for Germany can be downloaded via the DWD ftp Server. Tracks of SCS were calculated from DWD radar
data. The track data are not freely available, but can be provided upon request. ESWD severe weather reports are from the ESWD webpage
www.eswd.eu. ERA-Interim can be downloaded from the ECMWF sever. Front analyses can be provided upon request.

*Author contributions.* MK designed the research and wrote most parts of the paper. JW conducted the analyses of the environmental condi-
tions and wrote the corresponding sections. EF performed the SCS/hail track analyses, while SB combined the tracks with ESWD data and
frontal detections. SeS provided the data of synoptic fronts and wrote the corresponding section. SM and SeS edited the paper and provided
substantial comments and constructive suggestions for scientific clarification and further improvements.

*Competing interests.* The authors declare that they have no conflict of interest.

*Acknowledgements.* The authors thank the German Weather Service (DWD) and Meteo France for providing radar data, ESSL for making
available archived observations, and Siemens AG (S. Thern) for providing lightning data. ERA-Interim was downloaded from the ECMWF
web server. Data is stored at the Research Data Archive at the Karlsruhe Institute of Technology (KIT) and is available upon request to M.
Kunz.



**Table 1.** Number of HS events in the respective classes of maximum hail size diameter $D$ and track length $L$.

|  | $L < 50\,\text{km}$ | $L = 50\text{–}100\,\text{km}$ | $L > 100\,\text{km}$ |
|---|---|---|---|
| $D < 3\,\text{cm}$ | 311 | 98 | 64 |
| $D = 3\text{–}4.5\,\text{cm}$ | 190 | 102 | 72 |
| $D \geq 5\,\text{cm}$ | 63 | 35 | 50 |

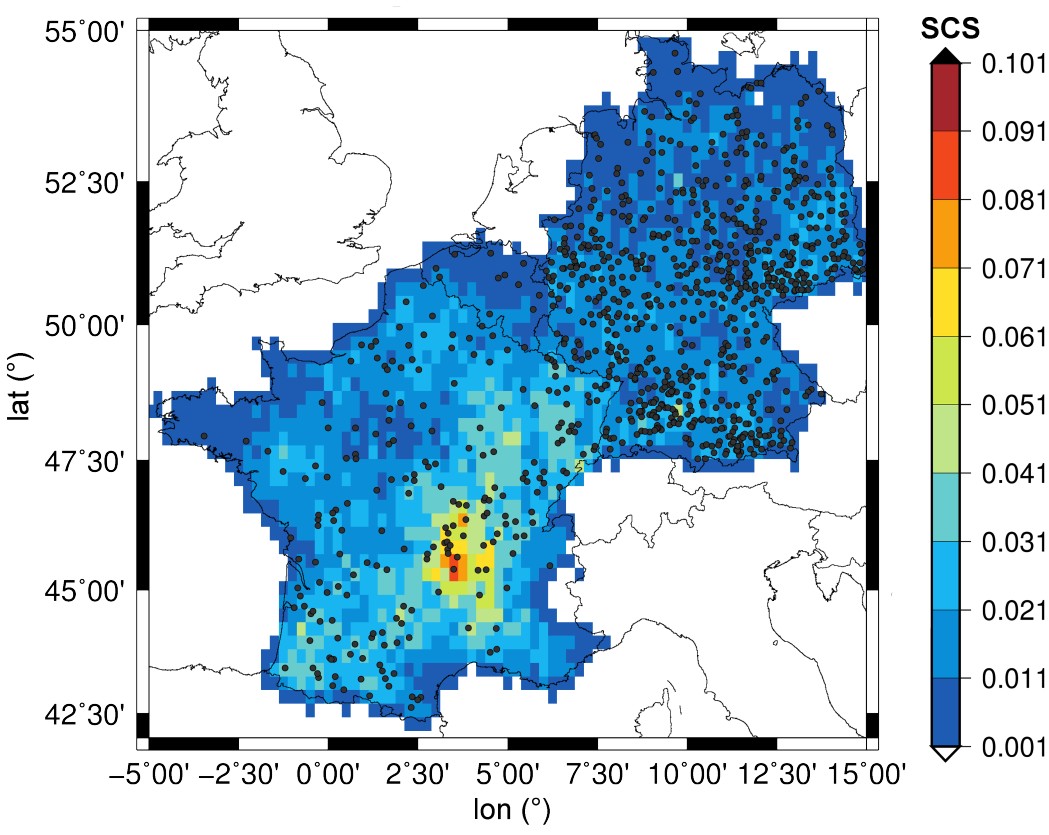

**Figure 1.** Number of SCS per year (center of each track) interpolated on a $0.25° \times 0.25°$ (color bar) and HS (black dots) between 2005 and 2014 over the investigation area (France, Germany, Belgium and Luxembourg).


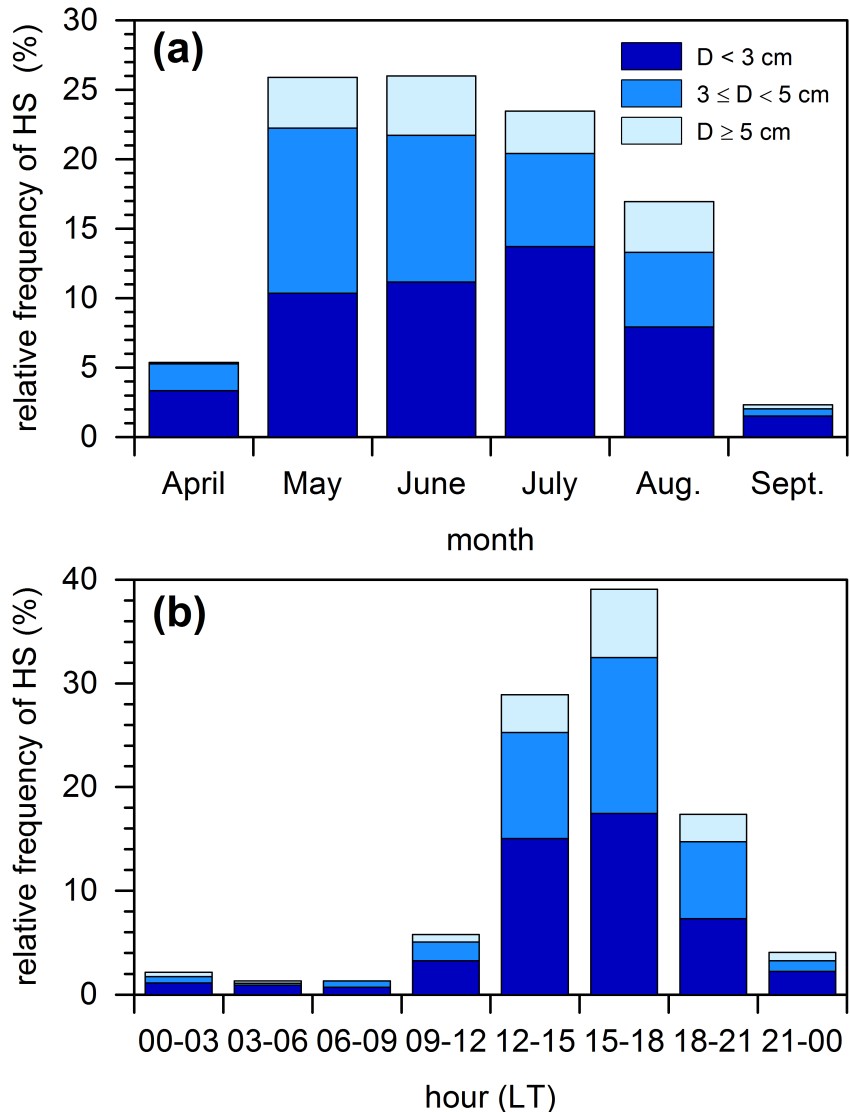

**Figure 2.** (a) Seasonal and (b) diurnal (local time LT) cycle of HS tracks (SHY, 2004–2014) depending on the hail-size diameter according to ESWD reports.
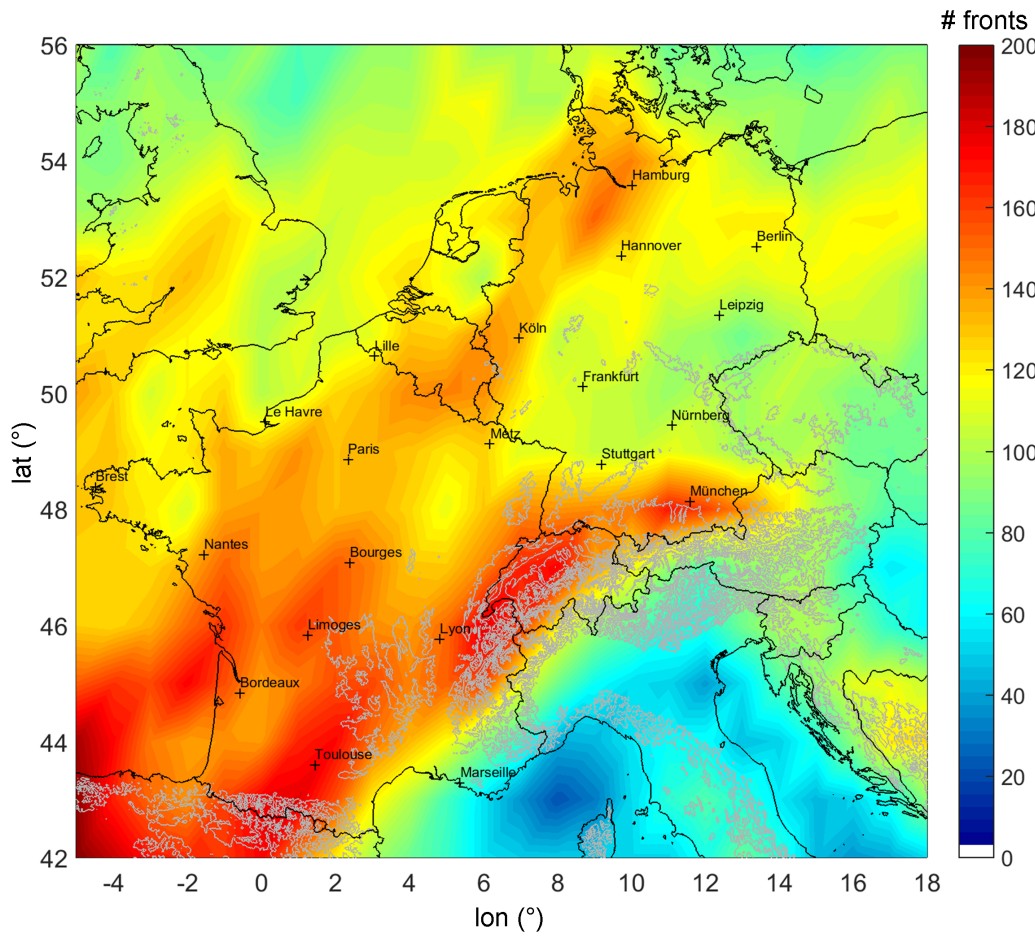

**Figure 3.** Number of synoptic-scale fronts per $1° \times 1°$ area between 2005 and 2014 (SHY) based on ERA-Interim reanalysis according to Schemm et al. (2015). Grey isolines represent the terrain (600, 1200, 1800, and 3600 m asl).

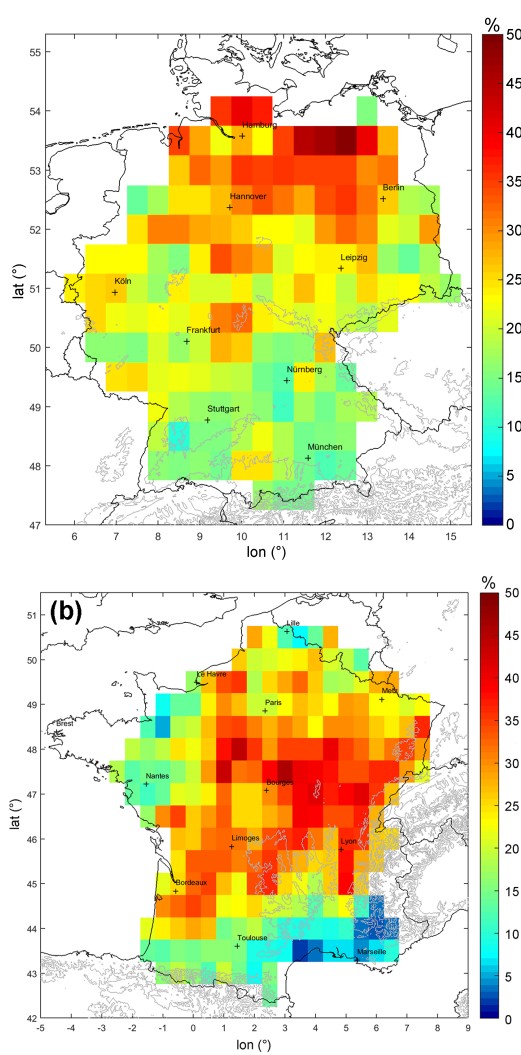

**Figure 4.** Share of frontal SCS (relative to all SCS; $r \leq 200\,\text{km}$) over (a) Germany and (b) France for $0.5° \times 0.5°$ (SHY, 2005–2014). Grid points containing less than 50 SCS tracks were left white.


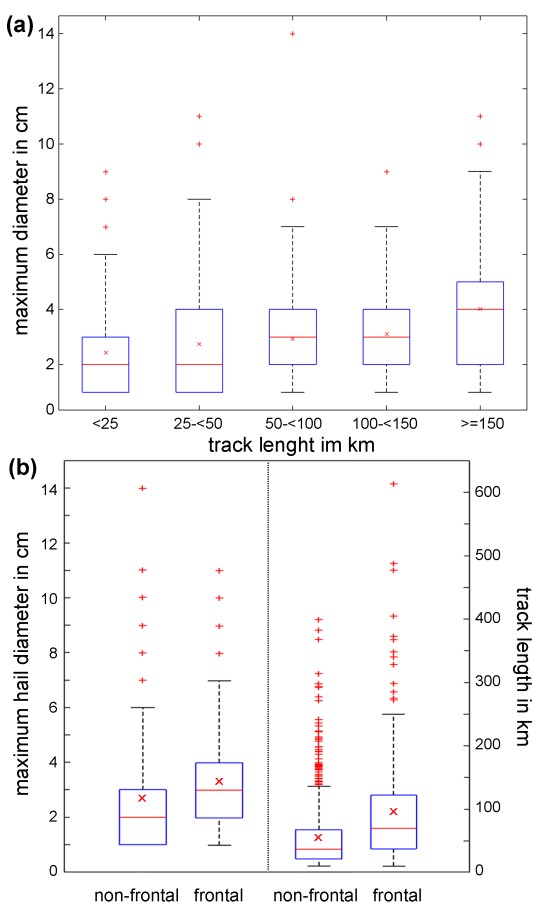

**Figure 5.** Boxplots showing (a) HS track lengths vs. maximum hail diameter according to ESWD reports and (b) maximum diameter (left part) and track length (right part) for HS events with/without a synoptic-scale cold front. Indicated in the boxplots are interquartile range (blue box); median and mean values (red line and red x); upper/lower 25% percentile ± interquartile range × 1.5 (black lines); data points outside of this range are marked as outliers (red crosses).


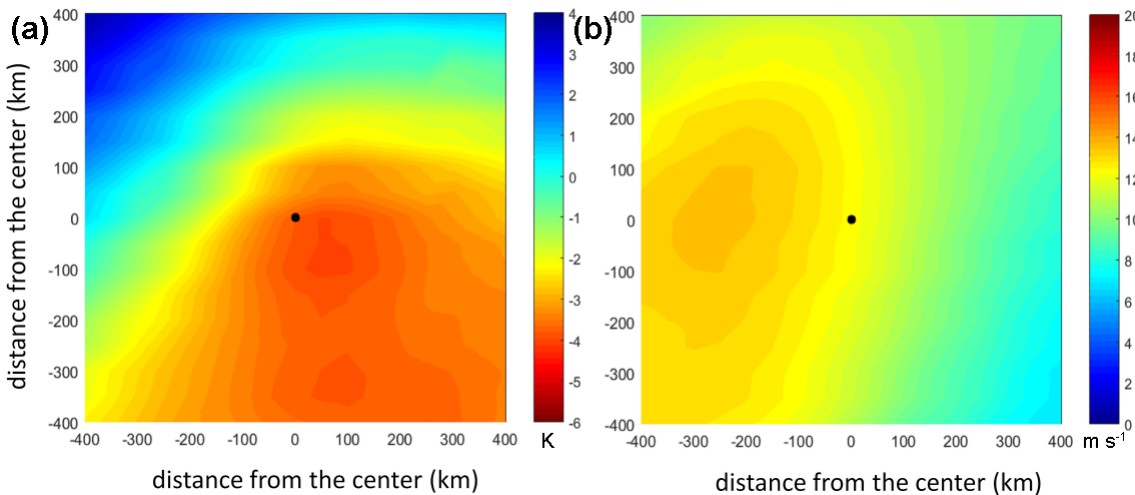

**Figure 6.** Composite analyses showing the average values of (a) SLI and (b) DLS from ERA-Interim in moving spatial windows centered at the track location (center) for all HS events between 2005 and 2014 (SHY; see Fig. 1).



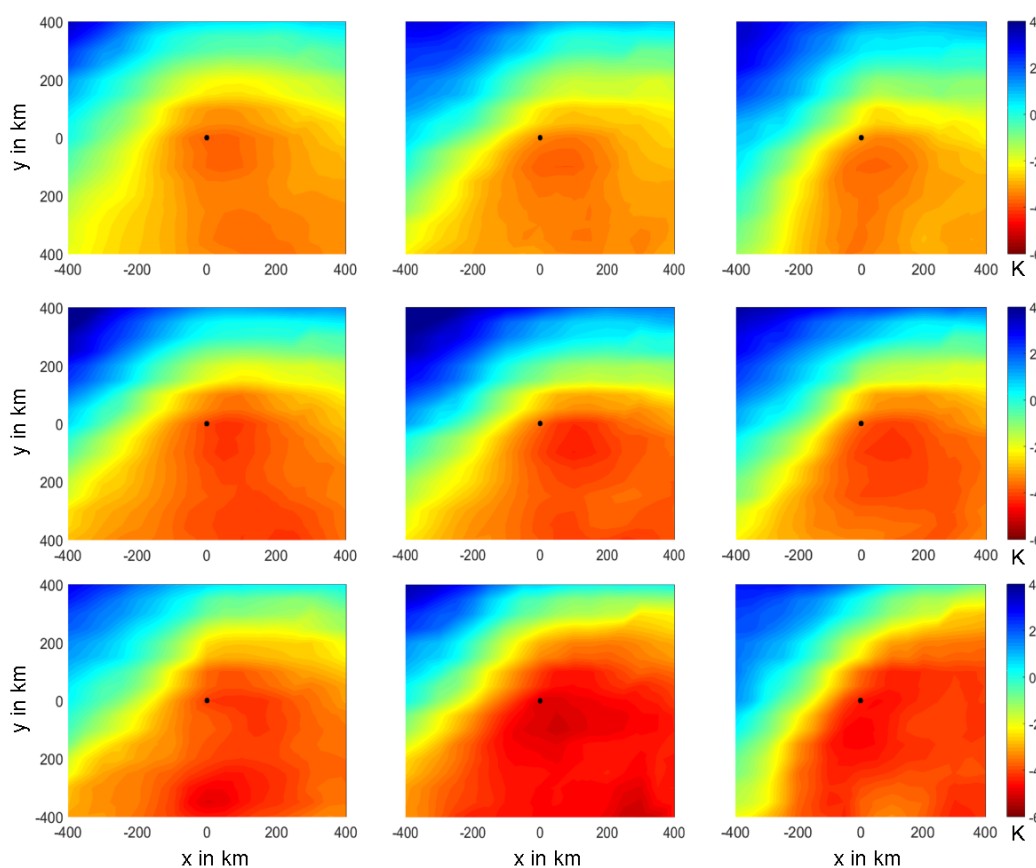

**Figure 7.** Composite analyses of SLI related to maximum observed hail diameters of $D < 3\,\mathrm{cm}$ (top row), 3–4.5 cm (middle row), and $\geq$ 5 cm (bottom row) and for track lengths of $L < 50\,\mathrm{km}$ (left column), 50–100 km (middle), and $\geq 100\,\mathrm{km}$ (right). The sizes of the subsamples are listed in Table 1.


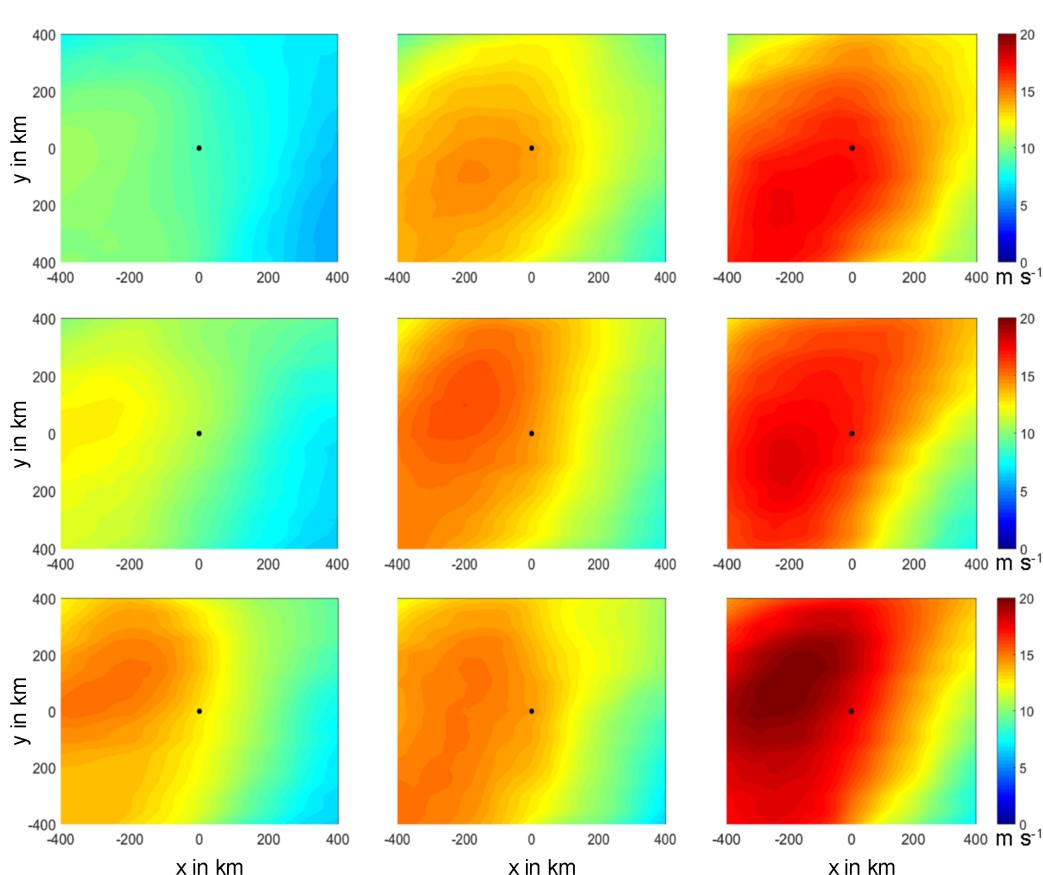

**Figure 8.** Same as Fig. 7, but for 0–500 hPa DLS.

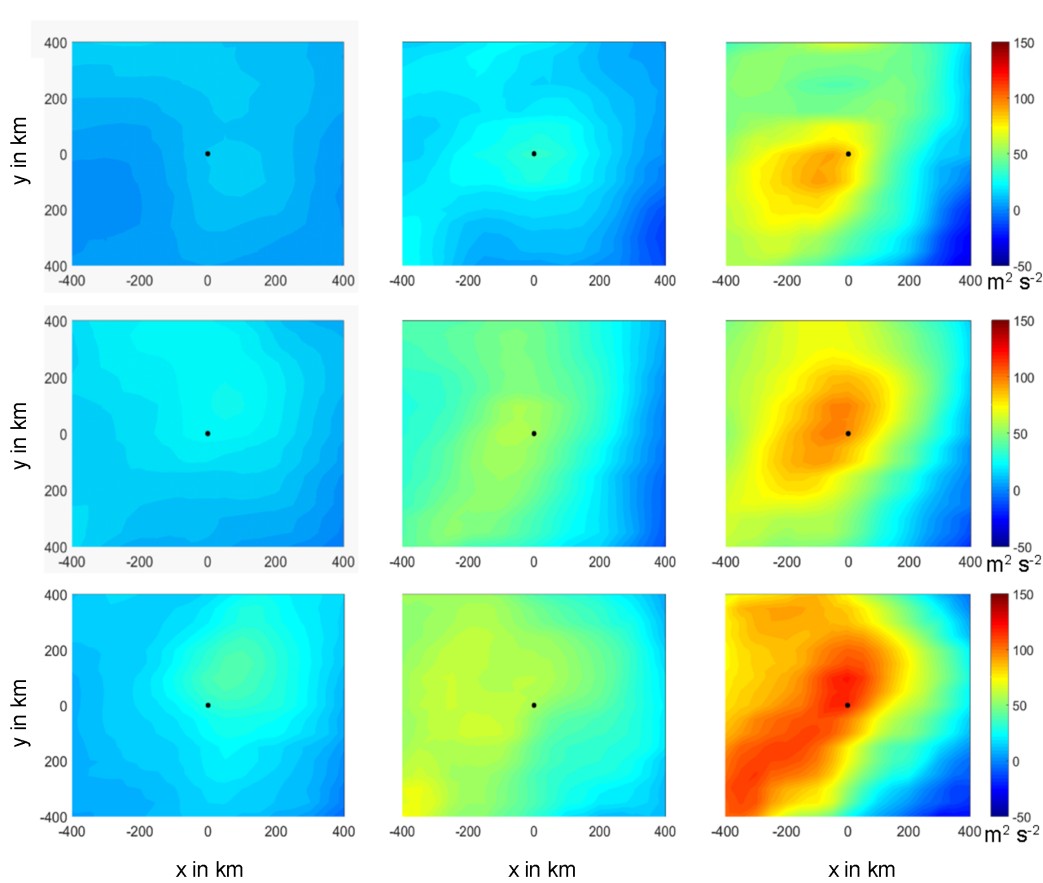

**Figure 9.** Same as Fig. 7, but for 0–3 km SRH.


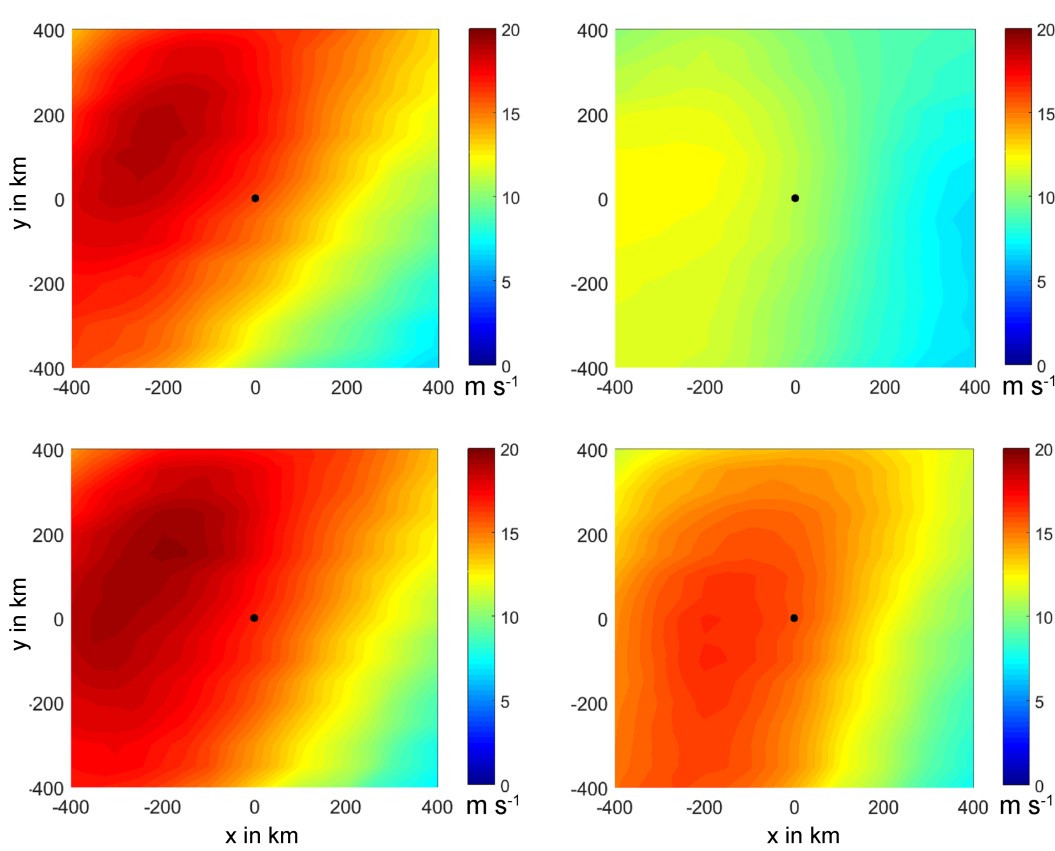

**Figure 10.** Composites of 0–6 km DLS for maximum observed hail diameters $D \geq 3$ cm and track lengths of $L < 75$ km (top row), and $L \geq 75$ km (bottom row) for frontal (left column) and non-frontal (right column) HS events.




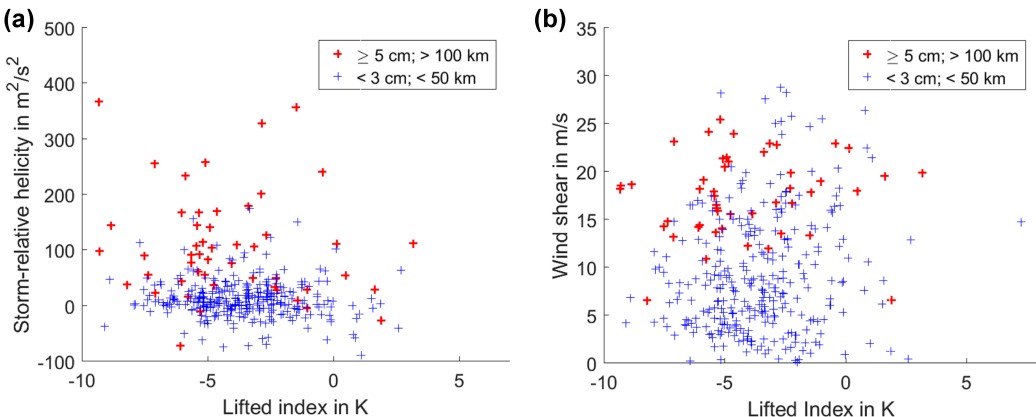

**Figure 11.** Scatter plots between (a) SLI and SRH and (b) DLS for two different categories of track length and hail diameter.


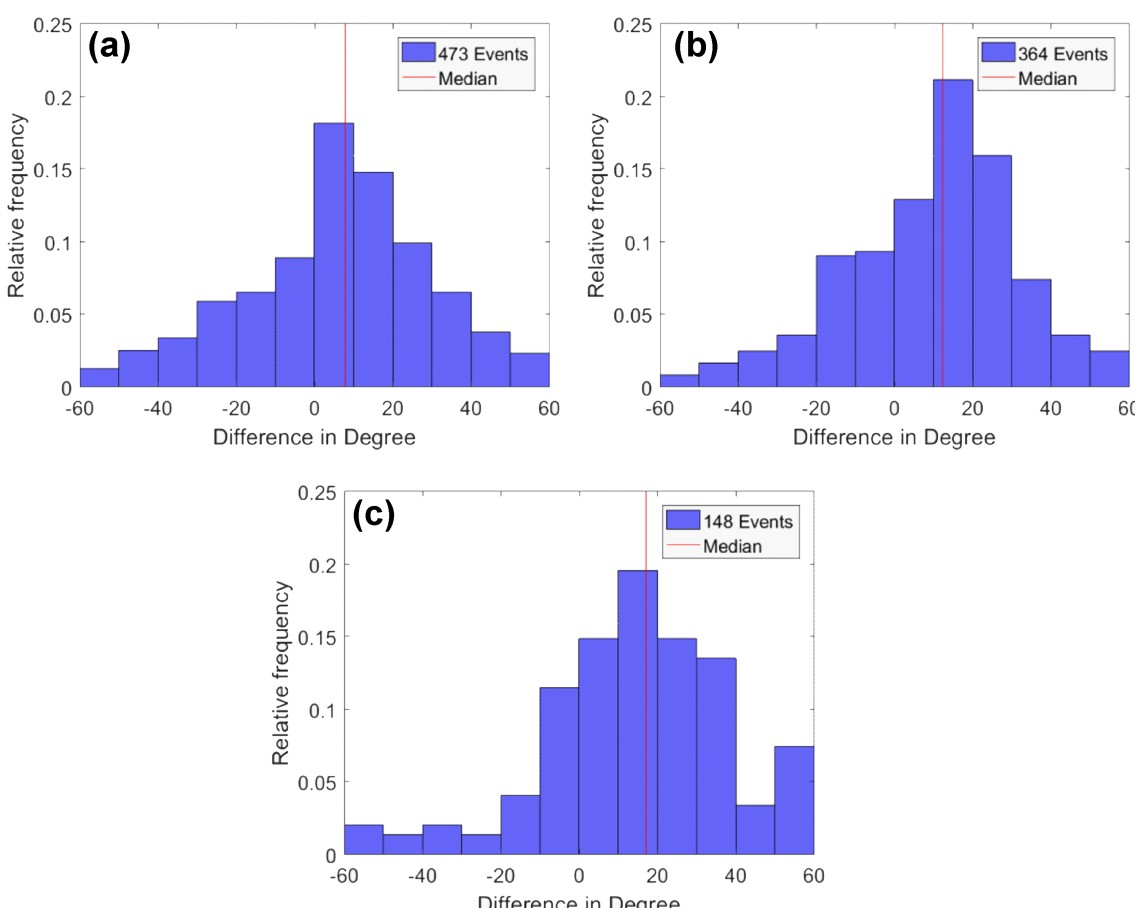

**Figure 12.** Histograms of HS events showing the relative frequency of the differences in the propagation direction between the storm motion vectors **c** and the wind in 500 hPa from ERA-Interim at the location and time of the HS events for three different diameter categories: (a) $D < 3$ cm, (b) $3 \leq D < 5$ cm, and (c) $D \geq 5$ cm. Median values are indicated by the red line.



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
