# Peer review of "Ambient conditions prevailing during hail events in central Europe"

_Natural Hazards and Earth System Sciences, 2019_

## Referee Comment (RC1) · Anonymous Referee #1 · 18 Feb 2020

**Review for**

**Ambient conditions prevailing during hail events in central Europe**

**by Kunz et al.**

**Summary:**

In their study, Kunz et al look at the ambient conditions during hail storm events in Central Europe. Different data sources are included in the analysis: Radar composites are used to track storm cells; the ESWD data base provides the hail diameter during storms; and the ERA-Interim reanalysis is used to characterize the environmental conditions at hail storm (HS) locations. The manuscript is well written, the scientific argumentation solid, and the balance of text and figures well motivated. Still, some aspects of the study should be addressed before it can be considered suitable for publication in NHESS. The major points are listed below (Major Concerns), some minor comments then follow in 'Minor Comments'.

**Major concerns:**

**1. Resolution of ERA-Interim:** Severe convective storms have a small spatial scale and can be triggered by small-scale topographic or other environmental features. How does this fit with the rather coarse spatial and temporal resolution of the ERA-Interim dataset? Of course, I notice that the authors are aware of this issue and they mention it at several places in the manuscript. Still, there remains the uncertainty whether the DLS (deep-level shear), SRH (storm-relative helicity) and SLI (surface lifted index) in ERA-Interim are really representative for the exact location where hail is recorded? I wonder whether

> - the authors should show some results of the high-resolution analysis data set that is mentioned in section 5.1, which however is not shon in the paper.

> - a limited time period (1 year?) could chosen and the analysis of the manuscript be repeated based on, e.g., high-resolution ECMWF analysis data.

I certainly don't expect the authors to repeat the whole analysis with higher-resolution data; but it would be nice to see how well the environmental conditions are really represented in ERA-Interim. I can imagine, for instance, that ERA-Interim does not capture the environmental conditions very well over the Massif Central, or over the Black Forest. I critical discussion, optimally complemented by a reduced analysis at higher resolution, should be included in the manuscript.

**2. Identification of fronts**: On P5,L118-120 some details about the front identification are given. The methodology to do so is well established and fine also for this study. Nevertheless, some aspects of the identification might be worthwhile to carefully address in the manuscript: (i) First, only cold fronts are considered, i.e. any convective storm associated with a warm front is not included? Is there a specific reason why the analysis is restricted to cold fronts? (ii) Stationary or slowly moving fronts (< 3 m/s) are not included in the attribution, based on the (fair) assumption that synoptic fronts associated with

extratropical cyclones typically move with reasonable speed? Still, the application of this criterion also might be a little artificial. (iii) Thermal fronts, or fronts associated with land-sea contrasts, are explicitly excluded. I guess that there are good reasons why to do that, but they should be more clearly discussed in the paper. (iv) Finally, pure geometric conditions are used (e.g., minimum length of 500 km) to separate 'significant' fronts from more 'fragmented' ones. I wonder, however, whether this also brings in some bias, because I can imagine that frontal fragmentation can be favored as the 'pure' and elongated synoptic fronts originating in the North Atlantic move in over Europe and move land inward. Frontal fragmentation might also be enhanced over orography, e.g. the Massif Central or the Black Forest. I am aware that objective identification of fronts is a difficult matter. But at least I would appreciate if the authors carefully address these limiting factors in the discussion.

**3. SCS vs HS; frontal vs. non-frontal:** The focus of this study clearly is on the HS, whereby these HS are classified according to their hail diameter based on the ESWD reports. That's fine, but brings in also some subjectivity. More specifically, how reliable/robust is the analysis if it is based on the maximum recorded hail size? Furthermore, Figure 1 also shows that the number of HS is small compared to the objectively identified SCS. Of course, I see that exactly these HS are at the heart of the manuscript's analysis. But, at some stage while reading the manuscript I 'defined' what I considered for myself as the most interesting part of the study. And, honestly, it is not the categorization according to the hail size and the track length. To me, the most interesting aspect is the distinction between frontal and non-frontal storms! I then wondered whether this would not be a better storyline than the existing one. Basically, the story would start with a climatology of the SCS over Central Europe; then, the next question addresss the fact whether the frontal tracks are associated with front or not. If this attribution is done, a next section could deal with the different characterization of the frontal vs the non-frontal storms. Here, the track length, the track duration, and also the presence of hail or not (including its size) could be discussed. Of course, I know that this storyline kind of reverses what the authors do now. One advantage of restructuring would be that the weakest part -- at least in my opinion --, i.e. the hail diameter, does not stand at the beginning, but only appears at the end of the chain. I would certainly not enforce the authors to reorder everything in the manuscript, but they might give it a thought.

**4. Application in forecasting:** This final major concern is a more 'philosophical' one. The basic aim of the study is to identify the severe and damaging storms, whereby the severity is (obviously) a combination of hail diameter and storm track length (footprint). These storms are taken as the basis of the analysis, and then the environmental conditions of these storms are considered. From a forecasting perspective, the reverse questions is much more interesting. In short, if we have an enviroment with this and that environmental conditions, what is the chance that we get a storm with a severe footprint? Please note that this reversed perspective is not exactly the same as the one studied in the manuscript, and this should be carefully discussed. Stated in a more practical way: How can the results of this study be used in an operational weather forecast? How would be the performance of such a forecasting algorithm if it were based on the environmental parameters used in this study? I think that the authors do a first step in this direction by quantitatively looking in P13,L378-383 at the predictive skill of certain parameter combinations. This part of the text should, however, be more detailed. The remained rather unclear to me how the predictive skill is determined...

**Minor comments:**

- P2,L24: 'large hail' → Does this refer to large hail stones, or does it refer to intense hail? It is not completely clear.

- P2,L30: What exactly is meant by 'footprint'?

- P2,L56: What is the temporal resolution of the cell tracking algorithm, more precisely: what is the resolution of the input fields?

- P2,L68: "Does the propagation direction of hailstorms depend on hail size?" → This question comes a little out of the blue, i.e. it was not motivated by the preceding paragraphs of the introduction. Is there a good reason to assume that the storm movement's direction depends on the hail size? If so, discuss this in the introduction.

- P4,L103: The SRH is calculated based on the storms movement (cx,cy), and based on the ERA-Interim

- P5,L141: "Several studies have provided evidence that suggests this threshold to provide sufficient skill for hail detection" -> Rephrase in clearer way.

- P5,L148: Here, the SCS tracks are first introduced. Possibly, it would be nice to learn already at this place about the typical characteristics of these tracks? For instance, about their length and duration? This is, in my view, reasonable because it allows already to set this characteristics to set into perspective of the limited spatial and temporal resolution of the ERA-Interim dataset.

- P6,L156: A minor detail: Wouldn't it be easier to take the spherical distance between the frontal points and the storm points instead of this (approximate) formula? It is approximate because the latitude actually changes along the storm track and along the front.

- P7,L211: "The frequency of both SCS and HS events shows a large spatial variability with an overall increase from north-to-south, i.e., with the distance to the Atlantic" -> I Am not sure whether I see this effect as clearly as it is stated in the text. There is a distance-to-Talantic effect near the coast, but further land inward it is less clear to me...

- P7,L212: From Figure 1 it is not clear that the regional hot spots are downstream of the Massif Central or the Black Forest. It is not clear because we don't see in this figure the direction of the wind.

- P9,L250: "not the individual fronts are done here" -> Sounds a little strange! It becomes clear with the succeeding sentence.

- P10,L187: "fronts create hail-conducive conditions mainly through two effects: along-front advection of moisture at lower levels leading to larger CAPE, and higher wind speed aloft enhancing vertical wind shear." -> What about the propagation of the front itself? This will also induce vertical lifting, as doers also the ageostrophic circulation associated with fronts? I would also assume that these effects depend on the intensity of the fronts. In short, a more comprehesnive discussion of the frontal effects would be nice -- as is actually done in the later discussion/conclusion sections of the paper.

- P10,L291: "fronts also largely determine the orientation of the tracks" -> Sounds a littel strange to me! Rephrase, e.g., to "the tracks are strongly coupled to the (typically eastward) propagation of the fronts.

- P12,L389: Based on the ERA-Interim winds, right-moving and left-moving supercells are distinguished. Can this reliably be done based on the coarse ERA-Interim winds? Note that the ERA-

Interim winds are representative for a approximately 80 x 80 km². The storm itself is, on the other hand, much smaller.

- P311,L314: "Considering the magnitude of DLS, it is found that the values are quite low with a mean of 12.5ms−1 around the HS events. Several authors have shown that organized convection 315 capable of producing larger hail develops only in sheared environments above around 10m m/s" -> Just for curiosity, myself not being an expert on storm dynamics. I would expect that there is an intermediate range of DLS that favors an intense and sustained convective activity. Too weak, but also too strong wind shear would hinder the convective storm development. It would be interesting to discuss this aspect somewhat in the manuscript, if my 'naive' thought is correct.

---

## Referee Comment (RC2) · Anonymous Referee #2 · 27 Feb 2020

The manuscript is well-written, correctly structured and methodologically sound. It lies within the scope of Natural Hazards and Earth System Sciences, as it considers the detection and modelling of a natural hazard (hail), with severe socioeconomic impacts. The authors present an empirical study with clear objectives, which involve the association of Severe Convective Storms (SCS) to fronts, the differences in the characteristics of frontal SCS differ to the non-frontal ones, the effect of environmental conditions to hail diameter and track lengths and weather the propagation direction of hailstorms is associated to hail size. Overall, the manuscript is of high quality and should be considered for publication in Natural Hazards and Earth System Sciences with few minor suggestions for revision presented below.

Figure 1: Consider splitting the Figure to two, as in Figure 4. In this manner, the spatial

[Figure]

patterns in Germany will be easier to detect and present (color scale).

L298: Seeing Figure 5, one might think 50-150 km class should be used instead. Is there another reason for choosing a 50-100km class?

L353-354: It would be less confusing to define the classes as D<3cm, D<75km and D>3cm, D>75km.

L356, L370: Figures 10 & 11 are the other way around.

L365-382: Perhaps consider removing this section. It does not contribute much to the findings and overall discussion and could be misleading. Statistical evaluation needs to be addressed also in terms of uncertainty analysis, which is not quantitatively presented in the manuscript.

---

## Author Comment (AC1) · 19 Mar 2020

The manuscript is well-written, correctly structured and methodologically sound. It lies within the scope of Natural Hazards and Earth System Sciences, as it considers the detection and modelling of a natural hazard (hail), with severe socioeconomic impacts. The authors present an empirical study with clear objectives, which involve the association of Severe Convective Storms (SCS) to fronts, the differences in the characteristics of frontal SCS differ to the non-frontal ones, the effect of environmental conditions to hail diameter and track lengths and weather the propagation direction of hailstorms is associated to hail size. Overall, the manuscript is of high quality and should be considered for publication in Natural Hazards and Earth System Sciences with few minor suggestions for revision presented below.

[Figure]

♯ We thank the reviewer very much for the time taken to review our manuscript and for the helpful and constructive comments. We are pleased that the reviewer approves the manuscript to be of high quality and well written. In preparing a revised manuscript, we will address all minor suggestions.

Figure 1: Consider splitting the Figure to two, as in Figure 4. In this manner, the spatial patterns in Germany will be easier to detect and present (color scale).
♯ The reviewer is right that the spatial pattern in Germany is difficult to detect. We will modify the color scale, but use the same scale for the entire area to allow direct comparison.

L298: Seeing Figure 5, one might think 50-150 km class should be used instead. Is there another reason for choosing a 50-100 km class?
♯ I fully agree, but the thresholds were set so that each sample contains at least 35 events. The classes as proposed would result in too small sample sizes for the long tracks (27 / 29 / 35 events for D $< 3$ / 3-4 / $\geq 5$ cm instead of 64 / 72 / 50 events). We will add a comment.

L353-354: It would be less confusing to define the classes as $D < 3$ cm, $D < 75$ km and $D > 3$ cm, $D > 75$ km
♯ We will consider this edit.

L356, L370: Figures 10 and 11 are the other way around.
♯ Thanks (this was a Latex compiler error)

L365-382: Perhaps consider removing this section. It does not contribute much to the findings and overall discussion and could be misleading. Statistical evaluation needs to be addressed also in terms of uncertainty analysis, which is not quantitatively presented in the manuscript.
♯ We will delete L378-382 at this paragraph do not give additional information. The reviewer is right, we did not consider uncertainty analysis, mainly because of the unreported HS events that are unknown.

---

## Author Comment (AC2) · 19 Mar 2020

In their study, Kunz et al look at the ambient conditions during hail storm events in Central Europe. Different data sources are included in the analysis: Radar composites are used to track storm cells; the ESWD data base provides the hail diameter during storms; and the ERA-Interim reanalysis is used to characterize the environmental conditions at hail storm (HS) locations. The manuscript is well written, the scientific argumentation solid, and the balance of text and figures well motivated. Still, some aspects of the study should be addressed before it can be considered suitable for publication in NHESS. The major points are listed below (Major Concerns), some minor comments then follow in 'Minor Comments'.

♯ We thank the reviewer very much for the time taken to review our manuscript and

for the helpful and constructive comments. We are pleased that the reviewer approves the manuscript to be well written with a solid scientific argumentation. In preparing a revised manuscript, we will address all major and minor suggestions.

Major Concerns
1. Resolution of ERA-Interim: Severe convective storms have a small spatial scale and can be triggered by small-scale topographic or other environmental features. How does this fit with the rather coarse spatial and temporal resolution of the ERA-Interim dataset? Of course, I notice that the authors are aware of this issue and they mention it at several places in the manuscript. Still, there remains the uncertainty whether the DLS (deep-level shear), SRH (storm-relative helicity) and SLI (surface lifted index) in ERA-Interim are really representative for the exact location where hail is recorded? I wonder whether - the authors should show some results of the high-resolution analysis data set that is mentioned in section 5.1, which however is not shown in the paper. - a limited time period (1 year?) could chosen and the analysis of the manuscript be repeated based on, e.g., high-resolution ECMWF analysis data. I certainly don't expect the authors to repeat the whole analysis with higher-resolution data; but it would be nice to see how well the environmental conditions are really represented in ERA-Interim. I can imagine, for instance, that ERA-Interim does not capture the environmental conditions very well over the Massif Central, or over the Black Forest. I critical discussion, optimally complemented by a reduced analysis at higher resolution, should be included in the manuscript.
♯ We will follow this suggestion and will include a short subsection (including an additional Figure; see last page) where we'll show and discuss exemplarily two quantities for higher resolved reanalysis data.
Note that reanalysis data is used only to estimate the convective environment, which has a much lower spatial variability compared to SCS probability. Storm's features are quantified from the tracking algorithm (also SRH). Triggering mechanisms are not considered as these are not reliably modelled even with high-resolution NWP models (see the bunch of literature on that topic).

In the meantime we have performed 7-years of hindcasts runs for Europe and Germany with a resolution of 7 and 2.8 km, respectively. Results will be shown in a next paper.

2. Identification of fronts: On P5,L118-120 some details about the front identification are given. The methodology to do so is well established and fine also for this study. Nevertheless, some aspects of the identification might be worthwhile to carefully address in the manuscript: (i) First, only cold fronts are considered, i.e. any convective storm associated with a warm front is not included? Is there a specific reason why the analysis is restricted to cold fronts? (ii) Stationary or slowly moving fronts ($< 3$ m/s) are not included in the attribution, based on the (fair) assumption that synoptic fronts associated with extratropical cyclones typically move with reasonable speed? Still, the application of this criterion also might be a little artificial. (iii) Thermal fronts, or fronts associated with land-sea contrasts, are explicitly excluded. I guess that there are good reasons why to do that, but they should be more clearly discussed in the paper. (iv) Finally, pure geometric conditions are used (e.g., minimum length of 500 km) to separate 'significant' fronts from more 'fragmented' ones. I wonder, however, whether this also brings in some bias, because I can imagine that frontal fragmentation can be favored as the 'pure' and elongated synoptic fronts originating in the North Atlantic move in over Europe and move land inward. Frontal fragmentation might also be enhanced over orography, e.g. the Massif Central or the Black Forest. I am aware that objective identification of fronts is a difficult matter. But at least I would appreciate if the authors carefully address these limiting factors in the discussion.

♯ (i) Warm-fronts may be relevant for setting of the precoventive conditions. But due to the slow ascending warm air in combination with warm-air advection aloft, they are not a relevant trigger for SCS. For this reason, we concentrate exclusively on cold fronts. We will add a comment on that. (ii-iii) Sea-breeze fronts and/or thermal boundaries, e.g. from Alpine pumping, are related and restricted to a specific territory. When considering a larger area as in our case, it does not make sense to mix the various types of fronts. We will add a comment (also about the thresholds) in the Data and

Method section and will include a short paragraph at the beginning of the Sect. 4 (SCS and fronts). (iv) The reviewer is right that our approach, in particular the fragmentation, brings in some bias. However, the criteria are necessary to filter out thermal contrasts that are not related to fronts with their lifting and cross-circulation. We will discuss that in the discussion section. (We think about a follow-up paper where we will consider all the above suggestions and separate the fronts according to different criteria/characteristics).

3. SCS vs HS; frontal vs. non-frontal: The focus of this study clearly is on the HS, whereby these HS are classified according to their hail diameter based on the ESWD reports. That's fine, but brings in also some subjectivity. More specifically, how reliable/robust is the analysis if it is based on the maximum recorded hail size? Furthermore, Figure 1 also shows that the number of HS is small compared to the objectively identified SCS. Of course, I see that exactly these HS are at the heart of the manuscript's analysis. But, at some stage while reading the manuscript I 'defined' what I considered for myself as the most interesting part of the study. And, honestly, it is not the categorization according to the hail size and the track length. To me, the most interesting aspect is the distinction between frontal and nonfrontal storms! I then wondered whether this would not be a better storyline than the existing one. Basically, the story would start with a climatology of the SCS over Central Europe; then, the next question addresss the fact whether the frontal tracks are associated with front or not. If this attribution is done, a next section could deal with the different characterization of the frontal vs the non-frontal storms. Here, the track length, the track duration, and also the presence of hail or not (including its size) could be discussed. Of course, I know that this storyline kind of reverses what the authors do now. One advantage of restructuring would be that the weakest part – at least in my opinion –, i.e. the hail diameter, does not stand at the beginning, but only appears at the end of the chain. I would certainly not enforce the authors to reorder everything in the manuscript, but they might give it a thought.

♯ (1) We considered not the mean, but the maximum hail diameter as this is usually

reported in the ESWD. We will clarify this. (2) Unfortunately not all hail events are reported in the ESWD (and not all SCS tracks are associated with large hail); reasons are an underreporting (particularly in France), but also hail diameters of less than 2 cm not included in the ESWD. We will include an explanation. (3) We focused on hailstorms and their properties because no study so far has investigated ambient conditions for different classes of hail size and track length. Our knowledge about hailstorms including their frequency and characteristics is still very limited. Note that in Germany but also in several other European regions/countries hail causes large economic losses and the highest share of insurance losses of all natural hazards. And most relevant for the damage potential of hailstorms are hailstone sizes and track length. (4) The idea of rearranging the manuscript is appealing if one focuses on the fronts. We have discussed this extensively with some of the co-authors. But as the focus should be still on hail for the reasons listed above, we will keep the previous order.

4. Application in forecasting: This final major concern is a more 'philosophical' one. The basic aim of the study is to identify the severe and damaging storms, whereby the severity is (obviously) a combination of hail diameter and storm track length (footprint). These storms are taken as the basis of the analysis, and then the environmental conditions of these storms are considered. From a forecasting perspective, the reverse questions is much more interesting. In short, if we have an environment with this and that environmental conditions, what is the chance that we get a storm with a severe footprint? Please note that this reversed perspective is not exactly the same as the one studied in the manuscript, and this should be carefully discussed. Stated in a more practical way: How can the results of this study be used in an operational weather forecast? How would be the performance of such a forecasting algorithm if it were based on the environmental parameters used in this study? I think that the authors do a first step in this direction by quantitatively looking in P13,L378-383 at the predictive skill of certain parameter combinations. This part of the text should, however, be more detailed. The remained rather unclear to me how the predictive skill is determined...

♯ This is indeed a very interesting point the reviewer raised highly relevant for forecasting / nowcasting. We are currently working on that topic in another project together with the German Weather Service DWD, where we include a subset of environmental conditions in the nowcasting algorithm to get an estimate about the expected future life time of the convective cells (a publication will follow). In case of hail events, however, it's not possible to reliably quantify the non-events because of a lack of comprehensive hail observations (see comments about the underreporting). We will discuss this issue in the conclusion section. After discussion with the co-authors, we decided to delete L378-383 about predictive skill because it gives not more information (except of skill score values) compared to the paragraph above.

Minor Comments
P2,L24: large hail → Does this refer to large hail stones, or does it refer to intense hail? It is not completely clear.
♯ This refers to large hailstones; we will change this sentence

P2,L30: What exactly is meant by footprint?
♯ The term footprint refers to the hail swath; we will add an explanation

P2,L56: What is the temporal resolution of the cell tracking algorithm, more precisely: what is the resolution of the input fields?
♯ The radar data, and thus the input fields, have a temporal resolution of 5 min (see P5, L123).

P2,L68: Does the propagation direction of hailstorms depend on hail size? → This question comes a little out of the blue, i.e. it was not motivated by the preceding paragraphs of the introduction. Is there a good reason to assume that the storm movement's direction depends on the hail size? If so, discuss this in the introduction.
♯ We agree and will delete this question.

P4,L103: The SRH is calculated based on the storms movement $(c_x, c_y)$, and based on the ERAInterim

♯ It may seem strange to mix up the input parameter. However, the tracking algorithm directly compute motion vectors for each of the detected cells, which are much more reliable compared to indirect methods such as those of Bunkers et al. (2000). This is the reason why we combined the two input parameters. We will modify this paragraph in order to clarify.

P5,L141: Several studies have provided evidence that suggests this threshold to provide sufficient skill for hail detection → Rephrase in clearer way.
♯ We will rephrase this sentence.

P5,L148: Here, the SCS tracks are first introduced. Possibly, it would be nice to learn already at this place about the typical characteristics of these tracks? For instance, about their length and duration? This is, in my view, reasonable because it allows already to set this characteristics to set into perspective of the limited spatial and temporal resolution of the ERA-Interim dataset.
♯ We will move all results from the Data and Methods section to the next section.

P6,L156: A minor detail: Wouldn't it be easier to take the spherical distance between the frontal points and the storm points instead of this (approximate) formula? It is approximate because the latitude actually changes along the storm track and along the front.
♯ That's almost the same. Changes in the latitude are considered by the cos term.

P7,L211: "The frequency of both SCS and HS events shows a large spatial variability with an overall increase from north-to-south, i.e., with the distance to the Atlantic" → I Am not sure whether I see this effect as clearly as it is stated in the text. There is a distance-to-Atlantic effect near the coast, but further land inward it is less clear to me...
P7,L212: From Figure 1 it is not clear that the regional hot spots are downstream of the Massif Central or the Black Forest. It is not clear because we don't see in this figure the direction of the wind.
♯ We will reformulate the whole paragraph.

P9,L250: not the individual fronts are done here → Sounds a little strange! It becomes clear with the succeeding sentence.
♯ We will modify this sentence

P10,L287: fronts create hail-conducive conditions mainly through two effects: along-front advection of moisture at lower levels leading to larger CAPE, and higher wind speed aloft enhancing vertical wind shear. → What about the propagation of the front itself? This will also induce vertical lifting, as doers also the ageostrophic circulation associated with fronts? I would also assume that these effects depend on the intensity of the fronts. In short, a more comprehesnive discussion of the frontal effects would be nice – as is actually done in the later discussion/conclusion sections of the paper.
♯ We will a little expand the discussion of the fronts in the discussion section, but delete the paragraph about the frontal modification of the environment in Sect. 4.2

P10,L291: fronts also largely determine the orientation of the tracks → Sounds a littel strange to me! Rephrase, e.g., to: the tracks are strongly coupled to the (typically eastward) propagation of the fronts.
♯ We will follow this suggestion

P12,L389: Based on the ERA-Interim winds, right-moving and left-moving supercells are distinguished. Can this reliably be done based on the coarse ERA-Interim winds? Note that the ERA- Interim winds are representative for a approximately 80 x 80 km$^2$ . The storm itself is, on the other hand, much smaller.
♯ Because of the storm's small-scale dimensions, we quantified the shift vector of the convective cells from the radar tracking algorithm. The wind field in 500 hPa, on the other hand, is modified only marginally by local-scale mechanisms but rather the result of the synoptic-scale setting of the pressure systems, which is sufficiently resolved by ERA-Interim. We will add a comment.

P11,L314: Considering the magnitude of DLS, it is found that the values are quite low with a mean of 12.5m/s around the HS events. Several authors have shown that

organized convection capable of producing larger hail develops only in sheared en-
vironments above around 10m/s → Just for curiosity, myself not being an expert on
storm dynamics. I would expect that there is an intermediate range of DLS that favors
an intense and sustained convective activity. Too weak, but also too strong wind shear
would hinder the convective storm development. It would be interesting to discuss this
aspect somewhat in the manuscript, if my naive thought is correct.
♯ Interesting. In the literature there is no upper limit reported. The findings of other
project of my group do not support the hypothesis of such an intermediate range of
DLS. Discussion of that point is a bit out of the context of this paper. – But I'll keep that
in mind for future research.
* * *
[Figure]

**Fig. 1.** Composites of temperature difference between 700 and 500∼hPa (LR; top row) and 0–6\,km shear (DLS; bottom row) for hail diameters $D \geq 5$\,cm and tack lengths $L \ge 100$\,km based on ERA-Interim

---

## Author Response (AR1)

We thank the two reviewers very much for the time taken to review our manuscript and for the helpful and constructive comments. We are pleased that the reviewers approved the manuscript to be well written with a solid scientific argumentation. In preparing a revised manuscript, we have considered all major and minor suggestions listed in the reviews.

**Reviewer #1**

In their study, Kunz et al look at the ambient conditions during hail storm events in Central Europe. Different data sources are included in the analysis: Radar composites are used to track storm cells; the ESWD data base provides the hail diameter during storms; and the ERA-Interim reanalysis is used to characterize the environmental conditions at hail storm (HS) locations. The manuscript is well written, the scientific argumentation solid, and the balance of text and figures well motivated. Still, some aspects of the study should be addressed before it can be considered suitable for publication in NHESS. The major points are listed below (Major Concerns), some minor comments then follow in 'Minor Comments'.

**Major Concerns**

1. Resolution of ERA-Interim: Severe convective storms have a small spatial scale and can be triggered by small-scale topographic or other environmental features. How does this fit with the rather coarse spatial and temporal resolution of the ERA-Interim dataset? Of course, I notice that the authors are aware of this issue and they mention it at several places in the manuscript. Still, there remains the uncertainty whether the DLS (deep-level shear), SRH (storm-relative helicity) and SLI (surface lifted index) in ERA-Interim are really representative for the exact location where hail is recorded? I wonder whether - the authors should show some results of the high-resolution analysis data set that is mentioned in section 5.1, which however is not shown in the paper. - a limited time period (1 year?) could chosen and the analysis of the manuscript be repeated based on, e.g., high-resolution ECMWF analysis data. I certainly don't expect the authors to repeat the whole analysis with higher-resolution data; but it would be nice to see how well the environmental conditions are really represented in ERA-Interim. I can imagine, for instance, that ERA-Interim does not capture the environmental conditions very well over the Massif Central, or over the Black Forest. I critical discussion, optimally complemented by a reduced analysis at higher resolution, should be included in the manuscript.

We followed this suggestion and included a new subsection (5.3 Effects of model resolution on convective parameters) where we discuss exemplarily two quantities – lapse rate and DLS – for CoastDat-3, a high-resolution reanalysis (~10 km, 1 hour resolutions) dynamically downscaled from ERA-Interim. The results, presented in the new Figure 11, show a larger variability of the fields, whereas the magnitude and the spatial distribution are almost unchanged.

Note that reanalysis is used only to estimate the convective environment, which has a much lower spatial variability compared to SCS probability. Storm's features are quantified from the tracking algorithm (also SRH). Triggering mechanisms are not considered as these are not reliably modelled even with high-resolution NWP models. (In the meantime we have performed 7-years hindcast runs using COSMO-EU and COSMO-DE for Europe and Germany with a resolution of 7 and 2.8 km, respectively. The results will be shown in a next paper.)

2. Identification of fronts: On P5,L118-120 some details about the front identification are given. The methodology to do so is well established and fine also for this study. Nevertheless, some aspects of the identification might be worthwhile to carefully address in the manuscript: (i) First, only cold fronts are considered, i.e. any convective storm associated with a warm front is not included? Is there a specific reason why the analysis is restricted to cold fronts? (ii) Stationary or slowly moving

fronts (< 3 m/s) are not included in the attribution, based on the (fair) assumption that synoptic fronts associated with extratropical cyclones typically move with reasonable speed? Still, the application of this criterion also might be a little artificial. (iii) Thermal fronts, or fronts associated with land-sea contrasts, are explicitly excluded. I guess that there are good reasons why to do that, but they should be more clearly discussed in the paper. (iv) Finally, pure geometric conditions are used (e.g., minimum length of 500 km) to separate 'significant' fronts from more 'fragmented' ones. I wonder, however, whether this also brings in some bias, because I can imagine that frontal fragmentation can be favored as the 'pure' and elongated synoptic fronts originating in the North Atlantic move in over Europe and move land inward. Frontal fragmentation might also be enhanced over orography, e.g. the Massif Central or the Black Forest. I am aware that objective identification of fronts is a difficult matter. But at least I would appreciate if the authors carefully address these limiting factors in the discussion.

We have expanded Section 2.3 Cold front detection and included some additional explanations in the results section. (i) Warm-fronts may be relevant for setting of the precoventive conditions. But due to the slow ascending warm air in combination with warm-air advection aloft, they are not relevant triggers for SCS. For this reason, we concentrate exclusively on cold fronts. We have added a comment at the beginning of Section 4. (ii-iii) Sea-breeze fronts and/or thermal boundaries, e.g. from Alpine pumping, are related and restricted to specific terrain characteristics. When considering a larger area as in our case, it does not make sense to mix the various types of fronts. We have added a comment (also about the thresholds) in the Section 2 Data and Method and included a short paragraph at the beginning of the Section 4 (SCS and fronts; L274-280). (iv) The reviewer is right that our approach, in particular the fragmentation, brings in some bias. However, the criteria are necessary to filter out thermal contrasts that are not related to fronts with their lifting and cross-circulation. This issue is now briefly discussed in the related paragraph of Section 6 Discussion. (We think about a follow-up paper where we will consider all the above suggestions and separate the fronts according to different criteria/characteristics).

3. SCS vs HS; frontal vs. non-frontal: The focus of this study clearly is on the HS, whereby these HS are classified according to their hail diameter based on the ESWD reports. That's fine, but brings in also some subjectivity. More specifically, how reliable/robust is the analysis if it is based on the maximum recorded hail size? Furthermore, Figure 1 also shows that the number of HS is small compared to the objectively identified SCS. Of course, I see that exactly these HS are at the heart of the manuscript's analysis. But, at some stage while reading the manuscript I 'defined' what I considered for myself as the most interesting part of the study. And, honestly, it is not the categorization according to the hail size and the track length. To me, the most interesting aspect is the distinction between frontal and nonfrontal storms! I then wondered whether this would not be a better storyline than the existing one. Basically, the story would start with a climatology of the SCS over Central Europe; then, the next question addresss the fact whether the frontal tracks are associated with front or not. If this attribution is done, a next section could deal with the different characterization of the frontal vs the non-frontal storms. Here, the track length, the track duration, and also the presence of hail or not (including its size) could be discussed. Of course, I know that this storyline kind of reverses what the authors do now. One advantage of restructuring would be that the weakest part -- at least in my opinion --, i.e. the hail diameter, does not stand at the beginning, but only appears at the end of the chain. I would certainly not enforce the authors to reorder everything in the manuscript, but they might give it a thought.

(1) We considered not the mean, but the maximum hail diameter as this is usually reported in the ESWD; added in Section 2.1. (2) Unfortunately not all hail events are reported in the ESWD (and not all SCS tracks are associated with large hail); reasons for this are a significant underreporting (particularly in France), but also small hail with diameters less than 2 cm that are not included in the

ESWD. We have added an explanation at the beginning of Section 3 (L233-239). (3) We focused on hailstorms and their properties because to the best of our knowledge no study so far has investigated ambient conditions for different classes of hail size and track length in Europe. Our knowledge about hailstorms including their frequency and characteristics is still very limited. Note that in Germany but also in several other European regions/countries hail causes large economic losses and the highest share of insurance losses of all natural hazards. Most relevant for the damage potential of hailstorms **are** hailstone sizes and track length. (4) The idea of rearranging the manuscript is appealing if one focuses on the fronts. We have discussed this extensively with some of the co-authors. But as the focus should be still on hail for the reasons listed above, we have decided to keep the previous order.

4. Application in forecasting: This final major concern is a more 'philosophical' one. The basic aim of the study is to identify the severe and damaging storms, whereby the severity is (obviously) a combination of hail diameter and storm track length (footprint). These storms are taken as the basis of the analysis, and then the environmental conditions of these storms are considered. From a forecasting perspective, the reverse questions is much more interesting. In short, if we have an environment with this and that environmental conditions, what is the chance that we get a storm with a severe footprint? Please note that this reversed perspective is not exactly the same as the one studied in the manuscript, and this should be carefully discussed. Stated in a more practical way: How can the results of this study be used in an operational weather forecast? How would be the performance of such a forecasting algorithm if it were based on the environmental parameters used in this study? I think that the authors do a first step in this direction by quantitatively looking in P13,L378-383 at the predictive skill of certain parameter combinations. This part of the text should, however, be more detailed. The remained rather unclear to me how the predictive skill is determined...

This is a very interesting point the reviewer raised and highly relevant for forecasting / nowcasting. We are currently working on that topic in another project together with the German Weather Service DWD, where we include a subset of environmental conditions in the nowcasting algorithm to get an estimate about the expected future life time of the convective cells (a publication will follow). In case of hail events, however, it's not possible to reliably quantify the probability of non-events because of a lack of comprehensive hail observations (see comments about the underreporting). We have added a paragraph in the Conclusion Section where we discuss this issue. After discussion with the co-authors, we have decided to delete L378-383 about predictive skill because it gives not more information (except of skill score values) compared to the paragraph above.

**Minor Comments**

P2,L24: large hail -> Does this refer to large hail stones, or does it refer to intense hail? It is not completely clear.

This refers to large hailstones; we have changed the sentence accordingly: "…large hail with a diameter of at least 2 cm…"

P2,L30: What exactly is meant by footprint?

The term footprint refers to the hail swath; we have added an explanation: "hail swath (envelope encompassing all hail streaks; footprint)"

P2,L56: What is the temporal resolution of the cell tracking algorithm, more precisely: what is the resolution of the input fields?

The input fields (radar data) have a temporal resolution of 5 min (see L138).

P2,L68: Does the propagation direction of hailstorms depend on hail size? →This question comes a little out of the blue, i.e. it was not motivated by the preceding paragraphs of the introduction. Is there a good reason to assume that the storm movement's direction depends on the hail size? If so, discuss this in the introduction.

We agree and deleted this question.

P4,L103: The SRH is calculated based on the storms movement ($c_x,c_y$), and based on the ERAInterim

It may seem strange to mix up the input parameter. However, the tracking algorithm directly compute the motion vector $\vec{c}$ for each of the detected cells, which are much more reliable compared to indirect methods such as that proposed by Bunkers et al. (2000). This is the reason why we combined the two input parameters. We have modify this paragraph (new L115-121) in order to clarify this.

P5,L141: Several studies have provided evidence that suggests this threshold to provide sufficient skill for hail detection → Rephrase in clearer way.

We have included a new sentence: "Several studies have provided evidence that this lower threshold is suitable to identify hail in radar data."

P5,L148: Here, the SCS tracks are first introduced. Possibly, it would be nice to learn already at this place about the typical characteristics of these tracks? For instance, about their length and duration? This is, in my view, reasonable because it allows already to set this characteristics to set into perspective of the limited spatial and temporal resolution of the ERA-Interim dataset.

To avoid any misinterpretation at the beginning of the manuscript, we have removed all results from the Data and Methods. Instead, we have included a new paragraph at the beginning of Section 3 that briefly summarizes the results of the data combination (L233-239).

P6,L156: A minor detail: Wouldn't it be easier to take the spherical distance between the frontal points and the storm points instead of this (approximate) formula? It is approximate because the latitude actually changes along the storm track and along the front.

That's almost the same. Changes in the latitude are considered by the cos term.

P7,L211: "The frequency of both SCS and HS events shows a large spatial variability with an overall increase from north-to-south, i.e., with the distance to the Atlantic" → I Am not sure whether I see this effect as clearly as it is stated in the text. There is a distance-to-Atlantic effect near the coast, but further land inward it is less clear to me... P7,L212: From Figure 1 it is not clear that the regional hot spots are downstream of the Massif Central or the Black Forest. It is not clear because we don't see in this figure the direction of the wind.

We have reformulated the whole paragraph (now L241-249).

P9,L250: not the individual fronts are done here → Sounds a little strange! It becomes clear with the succeeding sentence.

We have deleted this part because it's not necessary.

P10,L287: fronts create hail-conducive conditions mainly through two effects: along-front advection of moisture at lower levels leading to larger CAPE, and higher wind speed aloft enhancing vertical wind shear→ What about the propagation of the front itself? This will also induce vertical lifting, as doers also the ageostrophic circulation associated with fronts? I would also assume that these effects depend on the intensity of the fronts. In short, a more comprehesnive discussion of the frontal effects would be nice -- as is actually done in the later discussion/conclusion sections of the paper.

*We have expanded the discussion of the fronts in the discussion section and deleted the paragraph about the frontal modification of the environment in Sect. 4.2*

P10,L291: fronts also largely determine the orientation of the tracks → Sounds a little strange to me! Rephrase, e.g., to: the tracks are strongly coupled to the (typically eastward) propagation of the fronts.

*We changed this sentence accordingly. However, we have modified and moved the whole paragraph to the discussion section (now L526-542).*

P12,L389: Based on the ERA-Interim winds, right-moving and left-moving supercells are distinguished. Can this reliably be done based on the coarse ERA-Interim winds? Note that the ERA-Interim winds are representative for approximately 80 x 80 km². The storm itself is, on the other hand, much smaller.

*Because of the storm's small-scale dimensions, we quantified the shift vector of the convective cells from the radar tracking algorithm. The wind field in 500 hPa, on the other hand, is modified only marginally by local-scale mechanisms but rather the result of the synoptic-scale setting of the pressure systems, which is sufficiently resolved by ERA-Interim. We have add a comment (L455-458).*

P11,L314: Considering the magnitude of DLS, it is found that the values are quite low with a mean of 12.5m/s around the HS events. Several authors have shown that organized convection capable of producing larger hail develops only in sheared environments above around 10m/s →Just for curiosity, myself not being an expert on storm dynamics. I would expect that there is an intermediate range of DLS that favors an intense and sustained convective activity. Too weak, but also too strong wind shear would hinder the convective storm development. It would be interesting to discuss this aspect somewhat in the manuscript, if my naive thought is correct.

*Interesting. In the literature there is no upper limit reported. The findings of other project of my group do not support the hypothesis of such an intermediate range of DLS. As we haven't investigated this issue in our paper, we won't discuss this here. -- But I'll keep that in mind for future research.*

**Reviewer #2**

The manuscript is well-written, correctly structured and methodologically sound. It lies within the scope of Natural Hazards and Earth System Sciences, as it considers the detection and modelling of a natural hazard (hail), with severe socioeconomic impacts. The authors present an empirical study with clear objectives, which involve the association of Severe Convective Storms (SCS) to fronts, the differences in the characteristics of frontal SCS differ to the non-frontal ones, the effect of environmental conditions to hail diameter and track lengths and weather the propagation direction of hailstorms is associated to hail size. Overall, the manuscript is of high quality and should be considered for publication in Natural Hazards and Earth System Sciences with few minor suggestions for revision presented below.

Figure 1: Consider splitting the Figure to two, as in Figure 4. In this manner, the spatial patterns in Germany will be easier to detect and present (color scale).

The reviewer is right that the spatial pattern in Germany is difficult to detect. We have modified the color scale, but used the same scale for the entire area to allow direct comparison.

L298: Seeing Figure 5, one might think 50-150 km class should be used instead. Is there another reason for choosing a 50-100 km class?

I fully agree, but the thresholds were set so that each sample contains at least 35 events. The classes as proposed would result in too small sample sizes for the long tracks (27 / 29 / 35 events for D < 3 / 3-4 / ≥ 5 cm instead of 64 / 72 / 50 events). We have included a comment at the beginning of Section 5: "When defining the threshold values, it was taken into account that each class contains at least 50 events – except of the class L = 50–100 km and D ≥ 5 cm (Table 1). Using other thresholds, for example, 150 km instead of 100 km as suggested by the diameter–length relation shown in the boxplot (Figure 5) would result in too small sample sizes with less than 30 events."

L353-354: It would be less confusing to define the classes as D<3 cm, D<75 km and D>3 cm, D>75 km

Changed as suggested (L436-437)

L356, L370: Figures 10 and 11 are the other way around.

Thanks (this was a Latex compiler error)

L365-382: Perhaps consider removing this section. It does not contribute much to the findings and overall discussion and could be misleading. Statistical evaluation needs to be addressed also in terms of uncertainty analysis, which is not quantitatively presented in the manuscript.

The reviewer is right, we did not consider uncertainty analysis, mainly because of the unreported HS events that are unknown. We have therefore deleted the last paragraph of the statistical evaluation.

[revised manuscript text omitted]